# Chimpanzees produce diverse vocal sequences with ordered and recombinatorial properties

Cédric Girard-Buttoz [1,2,3,5 ✉], Emiliano Zaccarella [4,5], Tatiana Bortolato[1,2,3], Angela D. Friederici [4], Roman M. Wittig [1,2,3] & Catherine Crockford[1,2,3 ✉]

The origins of human language remains a major question in evolutionary science. Unique to human language is the capacity to flexibly recombine a limited sound set into words and hierarchical sequences, generating endlessly new sentences. In contrast, sequence production of other animals appears limited, stunting meaning generation potential. However, studies have rarely quantified flexibility and structure of vocal sequence production across the whole repertoire. Here, we used such an approach to examine the structure of vocal sequences in chimpanzees, known to combine calls used singly into longer sequences. Focusing on the structure of vocal sequences, we analysed 4826 recordings of 46 wild adult chimpanzees from Taï National Park. Chimpanzees produced 390 unique vocal sequences. Most vocal units emitted singly were also emitted in two-unit sequences (bigrams), which in turn were embedded into three-unit sequences (trigrams). Bigrams showed positional and transitional regularities within trigrams with certain bigrams predictably occurring in either head or tail positions in trigrams, and predictably co-occurring with specific other units. From a purely structural perspective, the capacity to organize single units into structured sequences offers a versatile system potentially suitable for expansive meaning generation. Further research must show to what extent these structural sequences signal predictable meanings.

[1] Institut des Sciences Cognitives Marc Jeannerod, CNRS, 67 Boulevard Pinel, 69675 BRON Lyon, France. [2] Taï Chimpanzee Project, Centre Suisse de Recherche Scientifique, Abidjan, Ivory Coast. [3] Department of Human Behaviour, Ecology and Culture, Max Planck Institute for Evolutionary Anthropology, 04103 Leipzig, Germany. [4] Department of Neuropsychology, Max Planck Institute for Human Cognitive Sciences, 04103 Leipzig, Germany. [5] These authors contributed equally: Cédric Girard-Buttoz, Emiliano Zaccarella. ✉email: cbuttoz@isc.cnrs.fr; crockford@isc.cnrs.fr

A major conundrum in evolutionary science has been reconstructing the evolution of language[1–5]. Given that language does not fossilize, a key line of research has been comparative, contrasting the communication systems of non-human animals (hereafter: animals) with that of humans. Unique to human language is its capacity to combine a limited sound set into words, and to combine words into rule-based hierarchically structured phrases, allowing the generation of endlessly new sentences and thereby new meanings. Animals, including non-human primates (hereafter: primates), typically use limited sound sets (e.g., primates[6], bats[7], and birds[8]), overlapping in size with human sound sets[9]. Thus, it is unlikely that, for animals, the size of the sound set is the factor limiting meaning generation. Rather, the animal capacity to generate communicative sequences is considered highly constrained, both in terms of structural systematicity and meaning generation[5]. In human language, words together with syntactic hierarchical structures permit flexible meaning construction. Such hierarchical structures have not been demonstrated in other animal communication systems although some animal calls show a limited capacity to encode meaning, and meaning can shift when single calls are emitted in short sequences[10].

Comparative studies with human and non-human primates have examined their capacities to process sequences both in perception and production. In terms of sequence perception, auditory discrimination tasks using familiarization-discrimination paradigms on artificially generated rule-based sequences have shown that some primates (cotton-top tamarins[11] and macaques[12]) appear to be sensitive to local sequence violations generated by artificial grammars although they fail to detect violations based on phrase structure relationships. This suggests that primates might possess some organizational principles applying over pattern regularities between neighboring units. Crucially, such positional principles appear to go beyond simple adjacency rules and include the capacity to detect sequence violations across non-adjacent relationships between distinct sounds[13,14]. Importantly, these tasks encode patterning but not meaning, as sounds are selected to be neutral and devoid of context. The observed behavioral ability could be based on a neuroanatomical structure which links the auditory cortex to the frontal cortex in non-human primates similar to humans[15]. This pathway has been discussed to support auditory-to-motor mapping in humans allowing vocal imitation of auditory input[16] and may at least serve a function of auditory pattern recognition in non-human primates.

With respect to production, there are no examples demonstrating that animals generate hierarchically structured vocal sequences with changes in meaning[5,17–20] where a sequence is broadly defined as the production of two or more different types of single vocal units within a short time of each other[21].

Due to methodological challenges[21], however, it cannot be ruled out that such examples exist in animals, nor likewise that intermediary examples of vocal sequence complexity exist. Establishing either structural flexibility or meaning content of even a single short vocal sequence is highly time-consuming. Therefore, animal studies have necessarily focused on a few vocal sequences per vocal repertoire but rarely offer quantitative assessment of sequence production across the whole vocal repertoire. Drawing definitive conclusions about the extent of animal capacities to generate sequences may thus be premature.

While both structure and meaning are crucial requirements for flexible meaning generation, a way to establish a link between the production of vocal sequences and sequence perception findings discussed above, is to assess whether vocal sequences in production follow structural rules, similar to rule-based sequence perception. Such structure of vocal sequences would facilitate the encoding of meaning, should meaning content be evident in the vocalizations. For example, if a repertoire has few vocal sequences, it cannot offer extensive meaning potential. Conversely, repertoires with many vocal sequences offer at least a greater potential for meaning generation. For instance, work on forest monkeys shows that the production of vocal sequences can be a source of new meaning generation whereby a sequence combining two alarm calls produce a new meaning related to travel[22]. This example illustrates that by combining meaning-bearing units (such as call types) into a sequence new meaning can be generated. However, this capacity seems limited to the combination of calls produced in alarm contexts in this species. Animal songs (e.g., whales[23] and song birds[24]) can contain hierarchical structuring of vocal sequences but without meaning content and so will not be discussed here. Hence, we postulate that the *structure* of a system that can encode flexible meaning should minimally, but not exclusively, require the following three structural capacities (summarized in Table 1):

1. *Flexibility:* are most sounds or calls in the vocal repertoire combined with most of the others (hereafter: single units)? Can single units, A, B, C, D, E… be combined as AB, AC, AD, AE, BC, BD, BE, CD, CE…?. If only few single units from the vocal repertoire occur in sequences (only AB, AD are options out of A, B, C, D, and E units), it is a limited system.

2. *Ordering:* Do single units within sequences follow ordering rules? i.e., both AB and BA can occur; however, if different ordering of the same single units should encode different information (such as coordination versus alarm), we would expect production biases at the structural level. Specifically, we expect that certain single units are more likely to be combined than others, and that they might occur in certain positions in the sequence depending on the units with which they are combined. If so then sequence production cannot be explained by random juxtaposition of single units.

3. *Recombination:* independently emitted short sequences are combined into longer sequences. Such recombination has the potential to increase meaning generation, overcoming limitations of a small vocal repertoire: e.g., independently produced bigrams (i.e., two independently used single units combined into a two-unit sequence) which can in turn be emitted in trigrams (sequences of three single units), and trigrams which can be found in longer sequences (e.g., AB can also occur as ABC, or separately emitted bigrams ABCD can occur as ABCD). Such recombination, might suggest some rudimentary capacity to treat independently emitted bigrams as units for more complex sequences. A vocal system that can flexibly combine all single units with

**Table 1 Three structural capacities that can expand the potential to encode flexible meaning generation.**

| Criterion | Example |
|---|---|
| 1. Flexibility: most single units in a repertoire combine with most other single units into sequences | A, B, C, … → [AB], [BC], [BA], [CB], … |
| 2. Ordering: ordered positioning of single units occur in sequences | [AB], [AC], [AD], [EF], [EG],… |
| 3. Recombination: short independently emitted sequences recombined into longer sequences | [[AB]C], [C[AB]], [[AB][CD]], … |

other single units but without the evident potential for some form of recombination, only provides limited meaning-generation potential.

Using a comparative perspective, we examine these three lower-level but universal capacities in human speech. These capacities loosely reflect a system that develops in early childhood as a pathway to hierarchical syntax: corpora studies show how words, initially produced in isolation, are flexibly assembled into two-word phrases, linearly ordered in the language of use, and then recombined into longer sequences[25,26].

Whether such structural capacities are evident in natural animal vocal production has not been comprehensively established: sequences reported so far in nonsinging species show limitations with respect to the three structural capacities laid out above.

For the first capacity, flexible combining of single units into sequences, across species vocal sequences occurring in long-distance and alarm calls use a minority of call types from the vocal repertoire, such that most call types are only ever emitted singly (e.g., chacma baboons[27], geladas[28], indris[29], Japanese tits[30]; but see Dahlin and Wright[31]). In alarm contexts, a few single units can be combined, changing the information about the predator type, urgency, or combining mobbing recruitment and warning calls into a sequence (e.g., several forest monkeys[22,32]; and Japanese tits[30]). However, the single units which are combined only represent a small portion of the total single-unit vocal repertoire of the species (but see Ouattarra et al.[33] for a possible exception). Furthermore, sequences emitted in an alarm context tend to occur as bigrams, where two independent calls are combined, and not in longer sequences ([10,34,35], but see refs. [30,33]).

For the second capacity, ordered production of single units within sequences, there are a few examples of ordering of single units produced in bigrams or longer sequences (e.g., Japanese tits[36]) which also demonstrate that changing unit order alters the information conveyed to conspecifics[30]. Longer sequences have shown variation in unit order for example only at the population level, indicating population differences (e.g., rock hyraxes[37]; birds[31]; and gibbons[38]).

For the third capacity, recombination of short independently emitted sequences, there are only a few reports of embedding of certain sequences within longer sequences. For instance, Japanese tits combine a common sequence ABC with a fourth element D to produce a sequence ABCD[30] and forest monkeys combine the sequence B_K with a third element H to form the longer sequence B_K_H[33]. However, reports of such phenomenon are rare and once again limited so far to a few single units per vocal repertoire.

In sum, we find only few reported examples for each of the three capacities laid out above, and no species other than humans, demonstrates all three structural capacities. This could result from researchers rarely evaluating the occurrence or flexibility of structured sequences across the whole vocal repertoire of nonsinging species. Hence it is quite possible that such a system that encodes the three structural capacities laid out above exists outside of humans.

Here, we examined vocal sequence production in the chimpanzee, a species known to produce vocal sequences across the whole vocal repertoire[39], but as yet lacking quantitative analyses of these sequences. We do not assess meaning in this study, but it is important to note that chimpanzee single-unit use can show high context-specificity across a relatively broad range of contexts compared to other species, including alarm, hunting, feeding and greeting[40] for a review see Crockford[41]. Chimpanzees also use single units within numerous vocal sequences, not only in their loud calls—pant hoot[42,43]—but in sequences occurring in contexts such as feeding, nesting, fusion, greeting, and travel[39,40,44].

The acoustic structure of calls emitted within a vocal sequence show acoustic fidelity to the calls emitted singularly[44]. However, a quantitative approach to examining vocal sequence patterning has not been conducted. These characteristics combined make this a suitable vocal system to examine the structure of vocal sequences in relation to the three structural capacities detailed above.

We aimed to quantify the structural properties of chimpanzee vocal sequences, assessing the potential for flexible meaning generation in chimpanzees. Thus, we tested the extent to which chimpanzees: (1) flexibly combine single-use calls into sequences, (2) produce ordered positioning of single units within sequences, and (3) recombine independently emitted sequences into longer sequences.

To reach this goal, we first extracted the use of single units as well as all the vocal sequences with two or more single units, using a full-vocal repertoire analysis. We tested each capacity as follows: (1) Flexibility: we first focused on bigrams (i.e., two-unit sequences). We constructed "networks" of bigrams to examine how extensively single units are combined with other single units across the vocal repertoire. (2) Ordering: we then assessed which bigrams were produced above-chance level (more frequently than by random juxtaposition of single units) and, focusing on these sequences, established whether single units were found consistently at specific positions (i.e., at the start or end of the sequence: positional bias), and were predictably preceded or followed by other specific single units (transitional bias). (3) Recombination: we focused on sequences with three single units (trigrams) and assessed which trigrams were produced above-chance level. We then established if frequently used bigrams were also reused within trigrams. If this occurred, we were also interested in whether the recombination of bigrams with a third single unit into trigrams followed some positional and transitional bias. We focused on sequences with two (bigrams) and three (trigrams) single units because they represent about 80% of the vocal sequences recorded in this study.

We used 900.8 h of data from 46 wild mature chimpanzees from Taï National Park, Ivory Coast, fully habituated to human observers, from three communities. We followed a systemic whole-repertoire approach where vocalizations were continuously recorded during focal animal sampling[27–29].

## Results

For this study, we used a call classification procedure, training listeners to classify calls according to sound and spectrographic information, resulting in high interrater reliability scores between coders. This method has proven to be the most accurate at classifying calls, especially in a noisy forest environment, as compared to semiautomatic classification of manually extracted metrics or fully automatic call classification algorithms[21]. To limit classification problems arising from the highly acoustically graded chimpanzee repertoire, we chose to classify broad call categories that show consensus across studies and chimpanzee populations (reviewed in Crockford[41]; Table 2, Supplementary Figs. 1 and 2) and the acoustic properties of which discriminate in cluster and discriminant function analyses[45]. Call types, such as grunt, hoo, bark, and scream, can be emitted singly (unpanted), or interdispersed with voiced inhalations (panted), producing a string of repetitions of alternations of pant + another vocalization (Supplementary Figs. 2 and 3). The use of unpanted or panted forms can result in contextual shifts. Single grunts, for example, are predominantly emitted at food, whereas panted grunts are predominantly emitted as a submissive greeting vocalization[41,42]. Single hoos are emitted to threats[46,47], but panted hoos are used in inter-party communication[48]. Unpanted and panted calls are

**Table 2 Discrimination of chimpanzee call types.**

| Call (unit) type | Abbreviation | Fundamental frequency (f0) | Shape of f0 | Noisiness | Single/panted | Occurrence in the repertoire |
|---|---|---|---|---|---|---|
| Grunt | GR | 70–700 Hz | Variable | Noisy | Single | 2087 (27.1%) |
| Panted grunt | PG | 100–200 Hz | Variable | Noisy | Panted | 853 (11.1%) |
| Hoo | HO | 200–700 Hz | Flat | Highly tonal | Single | 1561 (20.3%) |
| Panted hoo | PH | – | Flat | Highly tonal | Panted | 1015 (13.2%) |
| Bark | BK | 600–2000 Hz | Dome | Variable | Single | 337 (4.4%) |
| Panted bark | PB | 600–2000 Hz | Dome | Variable | Panted | 549 (7.1 %) |
| Pant | PN | 100–200 Hz (if visible) | Flat (if visible) | Noisy | Panted | 402 (5.2%) |
| Scream | SC | 800–2000 Hz | Flat midsection | Variable | Single | 312 (4.1%) |
| Panted scream | PS | 800–2000 Hz | Flat midsection | Variable | Panted | 317 (4.1%) |
| Non-vocal sounds (lip smack, raspberry) | NV | – | – | Noisy | Single | 184 (2.39%) |
| Whimper | WH | 350–1300 Hz | Flat | Highly tonal | Single | 78 (1.0%) |
| Panted roar | PR | 200–300 Hz, with bands below F0 (<100 Hz) | Flat | Noisy | Panted | 1 (0.01%) |

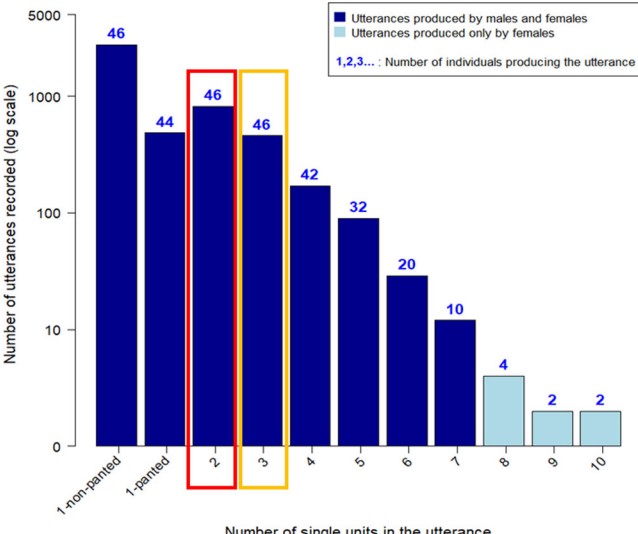

**Fig. 1 Frequency distribution of vocal sequences length.** The length refers to the number of different single units. The number of single units in the utterance is indicated on the *x* axis. The *y* axis depicts the number of recordings (log-transformed for ease of visualization). All utterances with one to seven single units (in dark blue) are produced by both sexes while utterances with eight or more single units (in light blue) are only produced by females. The number on top of each bar in blue indicates the number of individuals producing utterances of this particular length. The red box indicates the data used in the bigram analysis (Figs. 2 and 3) and the orange box indicates the data used in the trigram analysis (Fig. 4).

frequently combined into longer sequences[39]. Given these characteristics of panted calls, whilst panted calls may be sequences in their own right, for the purpose of this study we defined panted and unpanted form of calls as separate single units. Our rationale for treating panted calls as separate call types rather than as a sequence of e.g., grunt and pant, was motivated by previous studies demonstrating that pant-grunt and pant hoot are clearly stand-alone call types[42,48] (reviewed in Crockford[41]). For consistency in the treatment of panted call types, we include panted barks and panted screams as stand-alone call types. Please note that panted screams and panted barks are also reported across chimpanzee populations[41].

In the chimpanzee repertoire, variants of the same call, such as a hoo, are emitted in different contexts, such that the acoustic variants are context-specific, and elicit different behavioral responses during playback experiments from receivers[49]. However, it may be that hoos share a common function, such as to coordinate activities like resting and traveling[41,50]. Likewise, although screams are emitted in a range of contexts e.g., when being aggressed or during travel in pant-hoots, and some can be discriminated acoustically from others[51], they likely share an overarching function to recruit others[41,50]. Hence, for simplicity, in this study, we do not distinguish between different call variants but use a simplified classification whereby all hoos are classified as "hoo", all screams as "scream", and so on. This approach will, if anything, under-represent variation in vocal sequence use.

**Capacity 1: the flexible combination of single units within sequences.** To address capacity 1, the extent to which single-use calls are combined into sequences, we describe the portion of single units used in sequences and the length and diversity of sequences in the chimpanzee vocal repertoire. Chimpanzees produce 12 different types of single units (i.e., different call types, see Table 2 for the abbreviation of the name of these different call types as used in ""Results"). We analyzed 4826 utterances from 46 mature chimpanzees (Fig. 1), including adult and subadult males and females, older than ten years of age. In total, 3232 (67.2%) of these utterances contained only one single unit or repetitions of a single unit, e.g., only hoos or screams. Of these, 485, c.a. 10%, occurred in panted form e.g., panted hoos or panted screams (Fig. 1 and Supplementary Data 1). In total, 1584 (32.8%) utterances were sequences including 817 (16.9%) bigrams (i.e., sequences comprised of two units, where each unit also occurs singly, such as hoo + grunt), 458 (9.5%) trigrams (i.e., sequences comprised of three single units, where each unit also occurs singly such as hoo + panted hoos + panted scream) and 170 (3.5%), 90 (1.9%), 29 (0.6%), 12 (0.2%), 4 (0.1%), 2 (0.04%), and 2 (0.04%) sequences comprised of four, five, six, seven eight, nine and ten units respectively, where each unit also occurs singly (Fig. 1 and Supplementary Data 1). The length of a sequence was determined based on the succession of two or more single units which were each different from the preceding unit. However, the same single unit could be repeated in the sequence when separated by at least another single-unit type (e.g., we coded A_B_C_C as a trigram but A_C_B_C as a four-unit sequence). The rationale here is that the latter may also increase the potential for meaning generation. Using this criterion, we recorded 390 unique sequences. Please note that some studies also calculate sequence diversity such that each single unit only occurs once in the sequence. Using this alternative approach we found 282 unique sequences. For the rest of the result section, however, we report on sequences with

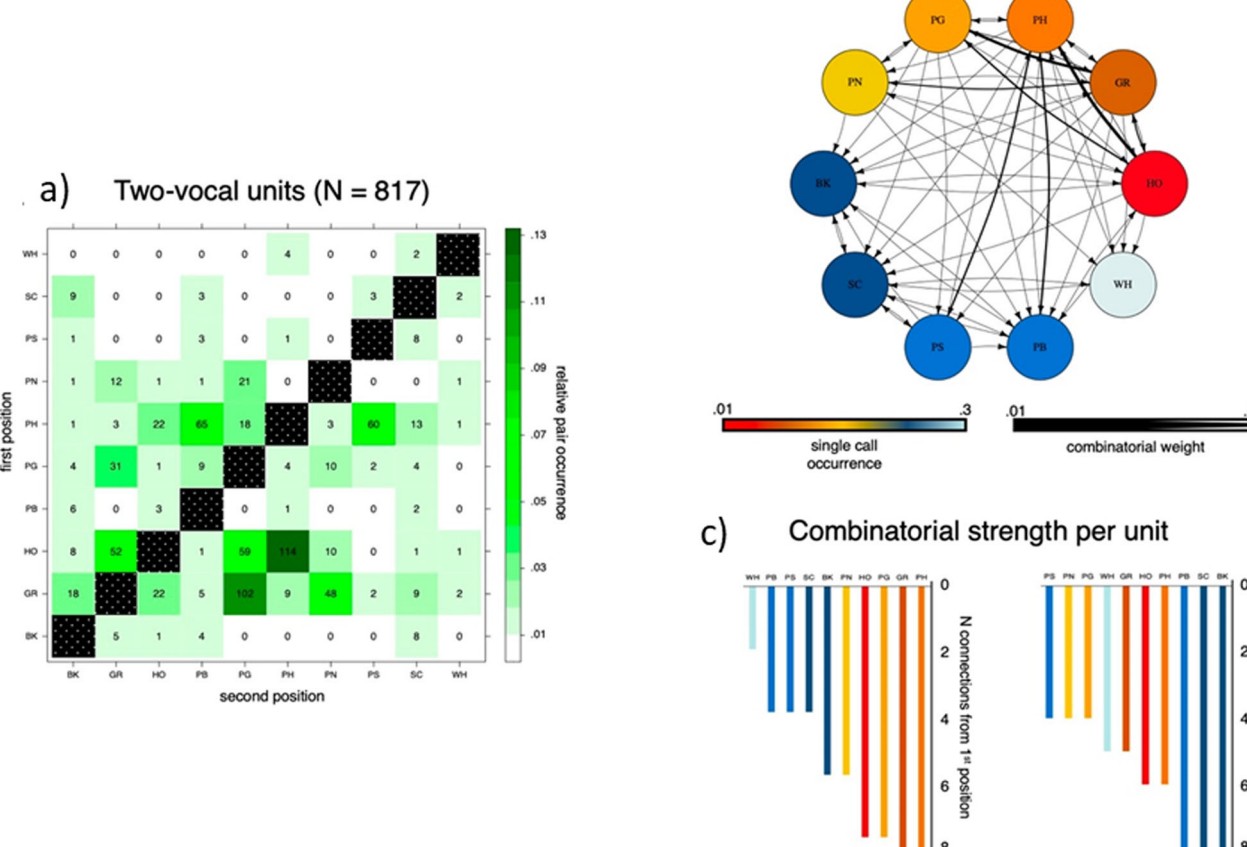

**Fig. 2 Bigram flexibility and ordering ($N = 817$ bigrams and 58 unique bigrams). a** Frequency distribution for two-unit vocal sequences (bigrams) with units occurring in first position listed along the y axis and units occurring in second position listed along the x axis. Color gradients (white-to-dark-green) represent the relative occurrence of each bigram within the two-units' set. In each cell, the absolute frequency count for each bigram is conversely reported. **b** The bigram combinatorial network with the ten single units depicted as circled nodes and color gradients (hot-to-cold) representing the number of times a certain unit is found in the bigram set. Note that the single units (NV) and (PR) that never occurred in bigrams are not shown here. This network was also used for the calculation of the Betweenness Centrality among the units. The size of the directional edges (arrows) expresses the number of times the specific bigram is found in the sample (thick-to-thin). **c** The number of different single units with which each single unit forms a bigram in the sample as first unit (left) or second unit in the bigram (right).

repeated single units in the sequence but not after itself (as described above). From an average of 13.2 focal hours per individual, all individuals produced trigrams. Two-thirds of the chimpanzees (32/46: 69.6%) produced sequences with five single units. In this sample, only females produced sequences with 8–10 single units. Examples of sequences produced by chimpanzees are provided in Supplementary Fig. 4.

Across our entire dataset, we found that all the 12 different types of single unit were used in sequences. PR only occurred once in this sample but within a sequence. From the 11 other single units, with the exception of non-vocal sounds (NV) which were combined with only four other single units, each of the 10 remaining single units were combined with 9 or 10 other different single units into sequences (e.g., across all sequences comprising GR, GR was combined into at least one sequence with BK and into at least one sequence, not necessarily the same one, with HO etc…). Four single units (GR, PB, PH, and PN, i.e., 33% of the single units) were combined with all other single units but PR across all sequences. Altogether this demonstrates tremendous flexibility in single unit combinatoriality.

When focusing on sequences with two single units (bigrams: $N = 817$), we found that chimpanzees produced 58 unique

bigrams (Supplementary Data 1). We represent all the bigrams in the dataset as a combinatorial network and show the corresponding combinatorial strength per unit (number of in/out connections) in Fig. 2b, c (see also Supplementary Fig. 5). Network metrics, and in particular betweenness centrality, revealed that PH is the single unit that is the most frequently combined with all other single units in the dataset ($PH_{Betweenness\ Centrality\ (BC)} = 12.45$). This suggests that PH might be a key single unit in the two-unit network and that it can be emitted flexibly in sequences with all other single units ($HO_{BC} = 4.566$, $SC_{BC} = 4.116$, $GR_{BC} = 3.816$, $BK_{BC} = 3.5$, $PB_{BC} = 2.533$, $PG_{BC} = 1.115$, $WH_{BC} = 1.033$, $PS_{BC} = 0.666$, $PN_{BC} = 0.2$).

**Capacity 2: ordered production of single units within sequences.** To address capacity 2, we focused on the 58 unique bigrams in the dataset and assessed (1) Positional bias: are the single units produced by chimpanzees biased towards the start or end position within a bigram? (2) Specific ordering frequency: do the bigrams show fixed order frequencies that go beyond random juxtaposition of single units? (3) Transitional bias: do relationships between these units exist when forming bigrams (i.e., the

**Table 3 Bayesian binomial test for positional, forward, and backward bias in two-unit sequences (bigrams).**

|  | Unit/ bigram | log(BF$_{10}$) | Posterior $\theta$ | 95% CI |
|---|---|---|---|---|
| Positional bias, bigrams | HO + $\diamond$ | 68.52 | 0.83 | 0.78–0.87 |
|  | GR + $\diamond$ | 18.72 | 0.68 | 0.62–0.73 |
|  | PH + $\diamond$ | 2.44 | 0.58 | 0.53–0.64 |
|  | $\diamond$ + PG | 34.03 | 0.75 | 0.70–0.80 |
|  | $\diamond$ + PB | 32.48 | 0.88 | 0.81–0.93 |
|  | $\diamond$ + PS | 18.37 | 0.82 | 0.74–0.90 |
|  | $\diamond$ + SC | 6.02 | 0.73 | 0.61–0.83 |
|  | $\diamond$ + BK | 5.77 | 0.72 | 0.61–0.82 |
|  | $\diamond$ + PN | 3.97 | 0.66 | 0.56–0.74 |
|  | $\diamond$ + WH | −0.88 | 0.60 | 0.50–0.79 |
| Forward bias, bigrams | HO→ PH* | −1.31 | 0.46 | 0.40–0.50 |
|  | PH→ PS* | 10.20 | 0.32 | 0.26–0.39 |
|  | PH→ PB* | 6.81 | 0.35 | 0.28–0.42 |
|  | GR→ PN* | 33.78 | 0.22 | 0.17–0.28 |
|  | GR→ PG* | −1.60 | 0.46 | 0.40–0.50 |
|  | HO→ PG* | 33 | 0.24 | 0.19–0.30 |
|  | PN→ PG | −0.81 | 0.59 | 0.50–0.72 |
| Backward bias, bigrams | HO ←PH | 35.76 | 0.85 | 0.79–0.90 |
|  | PH ←PS | 22.33 | 0.89 | 0.80–0.95 |
|  | PH ←PB | 7.19 | 0.71 | 0.61–0.80 |
|  | GR ←PN | 3.20 | 0.67 | 0.56–0.77 |
|  | GR ←PG | −2.18 | 0.53 | 0.50–0.59 |
|  | HO ←PG* | 15.50 | 0.29 | 0.24–0.36 |
|  | PN ←PG* | 69.22 | 0.10 | 0.07–0.15 |

Results from the Bayesian binomial test evaluating positional, forward, and backward bias in two-unit sequences (bigrams). For each analysis, we conducted Bayesian binomial tests on JASP[101] with default effect size priors and flat Beta (1,1) to quantify the relative likelihood for a potential positional or relationship bias within the sequence (see "Methods"). The number of successes over all trials for each unit was defined according to the number of occurrences found in the most frequent position (first or second position) for the positional analysis. The number of successes over all trials for each sequence was defined according to the number of times that υ was followed by ω (or that ω was preceded by υ), compared to the number of times that υ was followed by any other unit (or that ω was preceded by any other unit) for the two relationship analyses. Results are reported for the Bayesian factor log(BF$_{10}$), testing the hypothesis that the proportion of occurrences is higher than (unless specified) the default test value set at 0.5 (50%). Effect size estimates are reported as median posterior population ($\theta$) with Credible Intervals (CI) set at 95%[102]. *Testing the null hypothesis that the proportion of occurrences in the most frequent position is lower than the default test value set at 0.5. $\diamond$ = irrespective of unit type.

single unit υ always follows the single unit ω, or the single unit υ always precedes the single unit ω)?

*Positional bias.* The number of occurrences of every single unit in the first and second positions within each of the bigrams can be found in Fig. 2a (see also Supplementary Table 1). We used Bayesian binomial tests (see "Methods" and Table 3) over the number of occurrences of each unit in either the first or second position within a bigram. We found strong evidence for positional bias for nine out of the ten single units occurring in sequences. The single units HO, GR, and PH most reliably occurred in the first position in bigrams (Fig. 3a and Table 3). Conversely, PG, PB, PS, SC, BK, and PN showed an opposite bias toward the second position (Table 3). We found no bias for WH, which however only occurred 13 times in total.

*Ordering frequency.* We used a classic randomization procedure based on 1000 randomizations and we found that out of the 58 unique bigrams produced by the chimpanzees, 14 bigrams were produced above-chance level (i.e., more often than by random juxtaposition of single units from the repertoire, Fig. 3b). Out of the 14 above-chance bigrams, 7 were produced by at least 10 individuals (GR_PG, GR_PN, HO_PG, HO_PH, PH_PB, PH_PS, and PN_PG).

*Transitional bias.* We evaluated two different kinds of transitional relationships between the two single units forming the bigram, thus mimicking information-theoretic concepts used for quantifying processing efforts in the psycholinguistic literature[52–54]. Here, we focused on bigrams that occurred above-chance level and were produced by at least ten individuals. First, we tested how likely it was to find a unit ω in the second position, compared to all other possible follow-on units, given a unit υ in first position. We called this transitional possibility a forward relationship. Second, we tested how likely it was to find a unit υ in first position, compared to all the possible preceding units, given a unit ω in the second position. We called this alternative transitional possibility a backward relationship. None of the seven bigrams found above chance showed any forward relationship between the two single units within the sequence. Indeed, there was no unit in the first position that preferentially occurred with a certain unit in the second position (Fig. 3c and Supplementary Table 1). Thus, the first single unit in a bigram set no constraint on the following single unit. The second position did however appear to be bound to certain units in first position. We indeed found four bigrams with a strong backward bias—PH-PS, HO-PH, PH–PB, and GR-PN—suggesting that a certain second unit predominantly occurred when it was preceded by another specific unit (Fig. 3d and Table 3). For example, PS was in second position 67 times in a bigram and out of these 67 times it was preceded by a PH 60 times (89.5%) and only 7 times by other single units.

**Capacity 3: recombination of independently emitted bigrams into trigrams.** To address capacity (3), the extent to which recombination of independently emitted sequences occurs, we analyzed the structure of trigrams to explore how chimpanzees combine independently emitted bigrams with a third unit to produce trigrams. Of the 58 unique bigrams, 49 were also emitted with a third unit attached, as trigrams. The chimpanzees produced 458 trigrams, of which 104 were unique (Supplementary Data 1). We first assessed which trigrams were produced above-chance levels and which bigrams would appear above chance within trigrams. We thus first asked: (1) Specific ordering frequency: (a) do trigrams show nonrandom patterning beyond simple juxtaposition of single units? (b) Is there nonrandom positioning of bigrams within trigrams? (2) Positional bias: are bigrams biased toward a certain specific position within trigrams (head position—i.e., first and second position bias with a third unit taking the final position; tail position—i.e., second and third position bias with a third unit taking the head position)? (3) Transitional bias: do relationships between bigrams and specific third single units exist within trigrams (i.e., υω follows ε, or υω precedes ε)?

*Ordering frequency.* As for the analysis of bigrams above, we used classic randomization procedures and found that 49 unique trigrams were produced above-chance level (Supplementary Fig. 6). Out of these 49 above-chance trigrams, eight were produced by at least ten individuals (GR_PG_GR, GR_PG_PN, HO_PH_HO, HO_PH_PB, HO_PH_PS, PH_PB_PH, PH_PB_PS, and PH_PS_PB, Fig. 2b). Interestingly, four of the seven bigrams that were produced above-chance level and by at least 10 chimpanzees (GR_PG, HO_PH, PH_PB, PH_PS) are also found as part of these eight trigrams. When specifically considering all the bigrams occurring within trigrams, we identified 64 unique bigrams. Using the same randomization procedure as for the bigrams in sequences with two single units we found that 21 out of the 64 unique bigrams in trigrams were produced above the chance level (Fig. 4). Out of these 21 above-chance bigrams, 13

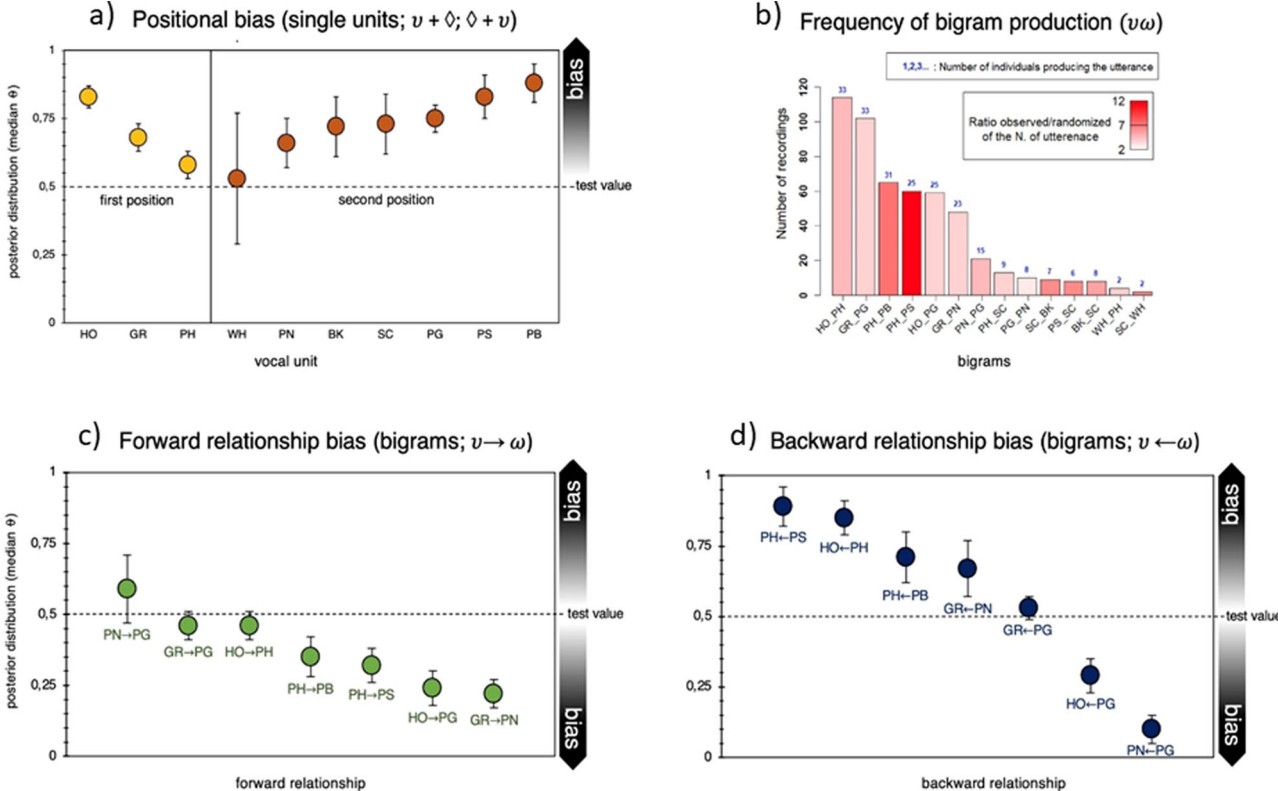

**Fig. 3 Positional bias, specific ordering frequency, and transitional bias for two-unit sequences (bigrams). a** Likelihood for a certain unit $v$ to occur in first position (yellow) or in second position (orange) in two-unit utterances, expressed as median posterior distribution ($\theta$). Bars stand for Credible Intervals (CI) at 95%. ◇ = irrespective of unit type (see Table 3). **b** Frequency of production of the bigrams that were produced above chance (i.e., >95% more likely than by random juxtaposition of single units). The height of each bar corresponds to the number of times each utterance was recorded. The color gradient in the bars depicts the number of times each utterance was observed divided by the number of times each utterance was present on average in each randomization (averaged over 1000 randomizations). The color ranges from the lowest ratio in white (i.e., the utterance was present in the observed data two times more than in the randomization) to the highest ratio in red (i.e., the utterance was present in the observed data 12 times more than in the randomized data). The number on top of each bar in blue indicates the number of individuals that produced each utterance. **c** Likelihood for a certain unit ($v$) in first position to be in a forward relationship (green) with a certain unit ($\omega$) in second position (see Table 3). **d** Likelihood for a certain unit ($\omega$) in second position to be in a feedback relationship (blue) with a certain unit ($v$) in first position, in two-unit utterances (see Table 3).

were produced by at least 10 individuals (GR_PG, HO_PH, PH_PB, PG_GR, PH_PS, PG_PB, PB_PH, PG_PN, PB_PS, PS_SC, SC_PS, PB_BK, and PS_PB). A comparison of the frequency distribution of bigrams in two-unit sequences with the frequency distribution of bigrams within trigrams reveals similar frequencies for the most frequent bigrams (Supplementary Fig. 7).

*Positional bias.* We used Bayesian binomial tests (see "Methods" and Table 4) to assess whether any of the four bigrams found above chance and emitted by at least 10 individuals at both two- and at three-unit level sequences (GR_PG, HO_PH, PH_PB, and PH_PS) showed bias to occur at head or tail position in the trigrams. We found that HO_PH, GR_PG, and PH_PB showed a strong positional bias towards the head position, while no effect was found for PH_PS (Fig. 4 and Table 4 and Supplementary Table 2).

*Transitional bias.* We tested for possible relationships between bigrams and single units in trigrams (i.e., [$v\omega$]$\varepsilon$ or $\varepsilon$[$v\omega$]). First, we asked in one case how likely is it to find $\varepsilon$, compared to all possible following units (forward relationship), given $v\omega$ in head position. Second, we asked how likely it is to find $\varepsilon$, compared to all the other possible preceding units (backward relationship), given $v\omega$ tail position. Transitional relationships between bigrams and single units revealed one strong forward relationship between

GR_PG and the following GR unit (Fig. 4 and Table 4), while the two-unit PH_PS was predominantly preceded by the HO unit (Figs. 4 and 5, Table 4, and Supplementary Tables 3 and 4).

**Literature review on sequence production in non-human primates.** We conducted a literature search on vocal sequences production in non-human primates (see details in the method section). We found reports of vocal sequence production in 31 non-human primate species and in all the four major taxa (i.e., apes and old-world monkeys (catarrhini): agile gibbons, *Hylobates agilis*[55], chimpanzees (our study), bonobos, *Pan paniscus*[56,57], gorillas, *Gorilla gorilla*[58], orang-utans, *Pongo pygmaeus*[59,60], blue monkeys, *Cercopithecus mitis*[61], Campbell's monkeys, *C. campbelli*[62], chacma baboon, *Papio ursinus*[27], DeBrazza's monkeys, *C. neglectus*[62], Diana monkeys, *C. diana*[63], geladas, *Theropitecus gelada*[64], olive baboons, *P. anubis*[27], putty-nosed monkeys, *C. nictitans*[22,65], red-capped mangabeys, *Cercocebus torquatus*[62], sooty mangabeys, *Cercocebus torquatus atys*[66], Thomas langurs, *Presbytis thomasi*[67], white-handed gibbons, *H. lar*; new-world monkeys (platyrrhini): golden lion tamarins, *Leonthopitecus rosalia*[68], common marmosets, *Callithrix jacchus*[69], Goeldi's marmosets, *Callimico goeldii*[70], pygmy marmosets, *Cebuella pygmaea*[71], silvery marmosets, *Mico argentatus*[72]; and prosimians (haplorrhini): Philippine tarsiers,

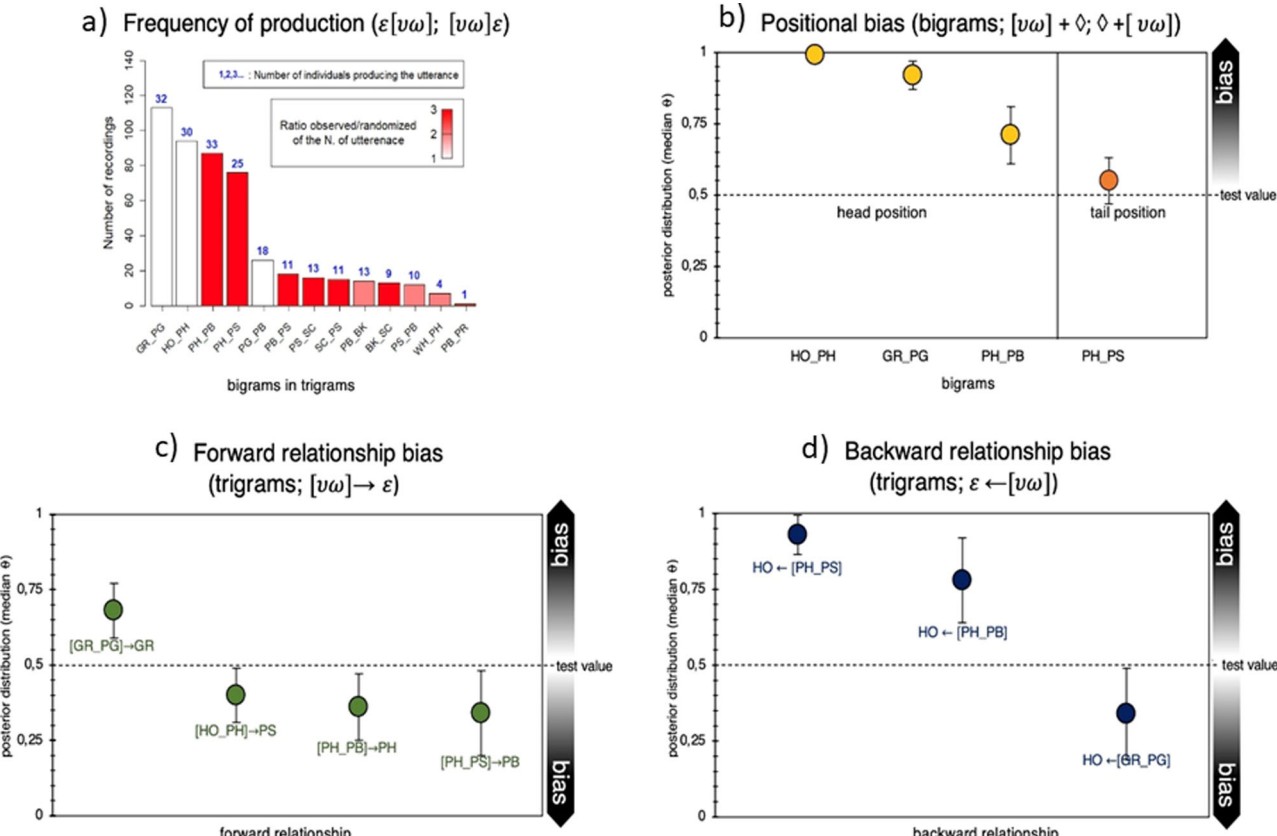

**Fig. 4 Specific ordering frequency, positional bias, and relationship bias for trigrams. a** Bigrams in trigrams that were produced above chance (i.e., >95% more likely than by random juxtaposition of single units). The height of each bar corresponds to the number of times each utterance was recorded. The color gradient in the bars depicts the number of times each utterance was observed divided by the number of times each utterance was present on average in each randomization (averaged over 1000 randomizations). The color ranges from the lowest ratio in white (i.e., the utterance was present in the observed data two times more than in the randomization) to the highest ratio in red (i.e., the utterance was present in the observed data 13 times more than in the randomized data). The number on top of each bar in blue indicates the number of individuals that produced each utterance. **b** Likelihood for a certain bigram $v\omega$ to occur in head position (yellow) or in tail position (red) in trigrams, expressed as median posterior distribution ($\theta$). Bars stand for Credible Intervals (CI) at 95%. $\diamond$ = irrespective of unit type (see Table 4). **c** Likelihood for a certain bigrams ($v\omega$) to be in a forward relationship (green) with a certain ($\varepsilon$) single unit following it (see Table 4). **d** Likelihood for a certain bigram ($v\omega$) to be in a backward relationship (blue) with a certain ($\varepsilon$) single unit following it (see Table 4). *The bigram PB_PR appeared in the dataset only 1 time and it appeared 250 times less in the randomizations as compared to the observed frequency (i.e., it appeared only 22 times in 1000 randomization so at an average frequency of 0.022), and not 3 times as indicated by the red color. We used a gradient of color from 1 to 3 to visualize the variation in the other bigrams bearing in mind that the ratio observed/randomized of the last bigram (PB_PR) is much higher than indicated in the figure.

*Tarsius syrichta*[73], common brown lemurs, *Eulemur fulvus*[74], indris, *Indri indri*[29], giant mouse lemurs, *Mirza mirza*[75], red-bellied lemurs, *Eulemur ribriventer*[74], crowned lemurs, *E. coronatus*[74], mangoose lemur, *E. mangoz*[76], Sahamalaza sportive lemur, *Lepilemur sahamalensis*[77]). At least one species of each of these taxa produced sequences comprising at least three different vocal units. In nonsinging species (i.e., species combining potentially meaning-bearing call types into vocal sequences) we could only find reports of a large diversity of different sequences (i.e., >10) in great ape species.

## Discussion

Compared to human language, the flexible use of vocal sequences in other species has been reported to be limited, although comprehensive quantitative analyses across whole animal vocal repertoires are rare. Here, we conducted a full quantitative analysis across the entire vocal repertoire to assess the usage of three structural capacities that should promote a flexible but ordered system, which would in principle provide a structural foundation for a versatile meaning generation.

First, we show that chimpanzees have a highly flexible vocal sequencing system, to an extent not yet reported for primate, as shown by our literature search (Fig. 6, see below). Taï chimpanzees produced 390 unique sequences comprising two or more single units (Fig. 1 and Supplementary Data 1). More than one-third of their vocal output includes at least two units, with 15% of vocal sequences containing three to ten units. Note these numbers are likely an underestimation of the vocal sequence potential of chimpanzees since the number of new sequences found had not reached asymptote after nearly 5000 recordings (Supplementary Fig. 8).

According to capacity (1), chimpanzees show a high degree of flexibility in vocal sequence output. 11 of the 12 vocalizations in the vocal repertoire were produced as single units and emitted in numerous different sequences with 4–9 other single units (Fig. 2b).

According to capacity (2), we found that the order in which single units were combined was flexible so that 52.6% of all bigrams were produced as AB and BA, at least once (Fig. 2b). In our dataset, we found 58 bigrams, 40 of which were 20 different pairs of single units produced with every single unit either as the

first or second element in the bigrams (e.g., a combination of A and B either as AB or BA) and 18 were bigrams which appeared only in a certain order (e.g., EF was produced but FE was never recorded). Yet single units followed some ordering properties when combined within sequences since we found evidence of positional and transitional bias using very conservative measures. At the two-unit level, we found a strong positional bias for almost every single unit, such that particular single units were preferentially emitted in the first and other particular units were emitted in the second position within bigrams (e.g., when combined with any other single unit, HO most predominantly occurs in first position, whereas PB predominantly occurs in last

**Table 4 Bayesian binomial test for positional, forward, and backward bias in three-unit sequences (trigrams).**

| | Bigrams/ trigrams | log(BF$_{+o}$) | Posterior $\theta$ | 95% CI |
|---|---|---|---|---|
| Positional bias, bigrams in trigrams | [HO + PH] + ◇ | 61.23 | 0.99 | 0.96–1 |
| | [GR + PG] + ◇ | 47.32 | 0.92 | 0.87–0.96 |
| | [PH + PB] + ◇ | 6.70 | 0.71 | 0.61–0.80 |
| | ◇ + [PH + PS] | −1.55 | 0.55 | 0.50–0.64 |
| Forward bias, bigrams in trigrams | [GR + PG] → GR | 5.93 | 0.68 | 0.59–0.77 |
| | [HO_PH] → PS* | 0.32 | 0.40 | 0.31–0.49 |
| | [PH_PB] → PH* | 1.43 | 0.36 | 0.25–0.47 |
| | [PH_PS] → PB* | 1.06 | 0.34 | 0.20–0.47 |
| Backward bias, bigrams in trigrams | HO ← [PH_PS] | 18.04 | 0.93 | 0.83–0.98 |
| | HO ← [PH_PB] | 3.88 | 0.78 | 0.61–0.91 |
| | HO ← [GR_PG]* | −0.27 | 0.34 | 0.13–0.49 |

Results from the Bayesian binomial test evaluating positional, forward, and backward bias of bigrams in trigrams. The same parameters as described in Table 2 were used (see "Methods"). The number of successes over all trials for each bigram was defined according to the number of occurrences found in the most frequent position (head vs. tail) in the trigrams for the positional analysis (positional bias). The number of successes over all trials for each utterance was thus defined according to the number of times that υω was followed by ε (or that υω was preceded by ε), compared to the number of times that υω was followed by any other units (or that υω was preceded by any other unit) for the relationship analyses (forward and backward bias). Results are reported for the Bayesian factor log(BF$_{10}$), testing the hypothesis that the proportion of occurrences is higher than (unless specified) the default test value set at 0.5 (50%). Effect size estimates are reported as median posterior population ($\theta$) with Credible Intervals (CI) set at 95%[102]. *Testing the null hypothesis that the proportion of occurrences in the most frequent position is lower than the default test value set at 0.5. ◇ = irrespective of unit type.

position). Second, there were several bigrams which were produced above-chance level (i.e., more than by random juxtaposition of single units). Third, we found a strong transitional bias within the above-chance bigrams, such that the single unit in second position was predominantly emitted with a specific single unit in first position (e.g., PH was preceded predominantly by HO). Worth noting, the fact that most bigrams can be produced with either unit emitted first, and that bigrams can also be preceded or followed by other units (Fig. 5 and Supplementary Fig. 5) suggests that the order in which the units are produced are not due to specific musculoskeletal constraints on the chimpanzee vocal articulatory system[78]. For example, several bigrams found in trigrams can be preceded or followed by four to eight different single units within the trigram (Supplementary Tables 3 and 4 and Fig. 5). Likewise, within bigrams, although we found ordering effects, we found occurrences of most bigrams being emitted with either element first (Fig. 2a, b).

To address capacity (3), recombination, we conducted the same analyses at the three-unit level (trigrams). This showed that close to half of the unique trigrams (47%, 49 out of 104) were produced above-chance level. We found that bigrams showed the same order preference whether emitted as a bigram or within a trigram (e.g., HO_PH occurs as a bigram but can also occur with an added third unit in a trigram: HO_PH_PB or HO-PH_PS). We also found strong positional bias, such that certain bigrams predictably occurred in either head or tail positions within trigrams (e.g., third units added to GR_PG typically occur after and not before the bigram is emitted: GR_PG_X). Finally, we found a strong transitional bias of bigrams within trigrams, such that some bigrams in the head/tail position were emitted with the highest likelihood with a certain specific unit in tail/head position (e.g., the bigram GR_PG is more likely to be followed by GR than by any other single unit while the bigram PH_PB is more likely to be preceded by HO than by any other single unit). Our results also indicate that some bigrams are reused as ready-combined calls into trigrams. This might offer intriguing evidence for a potential pairing prerequisite during vocal communication in animals[30,79]. If these single units carry meaning, it may be that chimpanzees form combined meanings from two single units, which can, in turn, be recombined with a third unit to eventually output a third combined meaning. However, without looking into

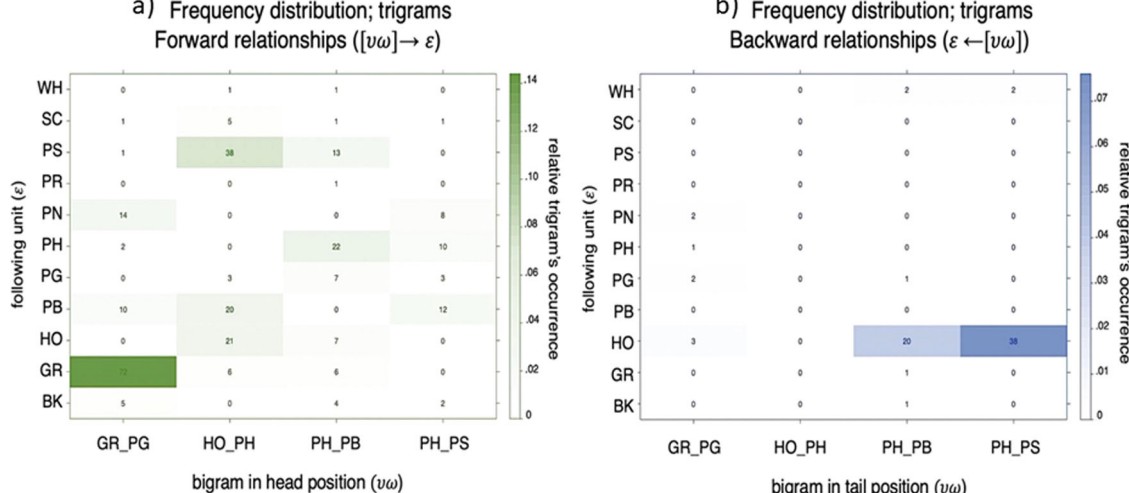

**Fig. 5 Frequency distribution for bigrams within trigrams. a** Bigrams occurring in head position (υω) along the x axis and subsequent units (ε) along the y axis. Color gradients (white-to-green) represent the relative occurrence of each bigram within the trigram' set. In each cell, the absolute frequency count for each three-unit sequence is also reported. **b** Frequency distribution for trigrams with bigrams occurring in tail position (υω) along the x axis and preceding units (ε) along the y axis. Color gradients (white-to-blue) represent the relative occurrence of each bigram within the trigram' set. In each cell, the absolute frequency count for each trigrams is also reported.

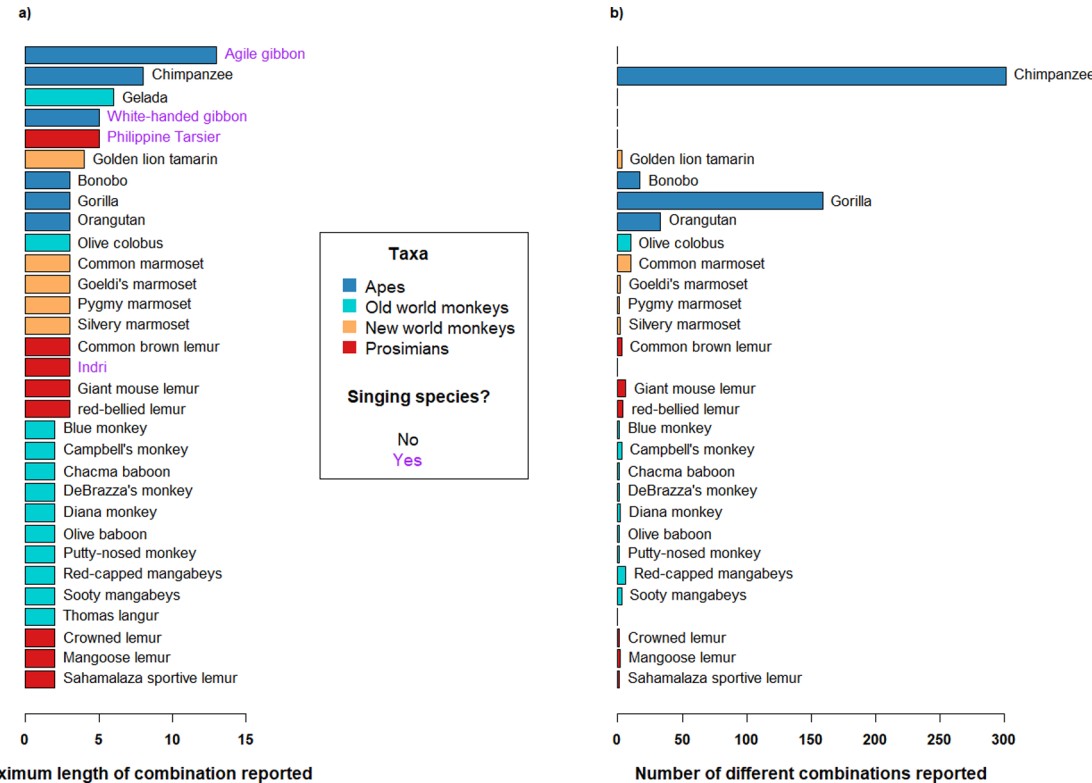

**Fig. 6 A literature review comparing vocal sequencing across primates.** The barplot on the left indicates the maximum length of sequences reported in each species (i.e., the number of unique single units emitted in a single sequence) (**a**). The barplot on the right indicates the number of different sequences reported to comprise at least two different single units (**b**). We counted sequences in which different single units occurred at least once. If the same single unit was repeated later in the sequence this was not counted as a new sequence. Species showing no number of different sequences may reflect lack of published information. The color of the bars in both plots indicate the taxa of the species. Apes (including great and lesser apes), old-world monkeys, new-world monkeys, and prosimians are depicted in dark blue, light blue, orange, and red respectively. The singing species are indicated by the purple text. Singing species are species such as Indris or gibbons which emit different single units in song. These units are not necessarily emitted singularly nor in other parts of the vocal repertoire. Superscript numbers refer to the relevant citations listed below.

contextual information, which goes beyond the scope of the present study, our analysis cannot exclude that the pairing effects in the three-unit vocal sequence result from simple transitional relationships between adjacent elements. In this respect, trigrams containing bigrams produced above chance, like HO←[PH–PS] or HO←[PH–PB], might be rather analyzed as HO←PH←PS or HO←PH←PB, where single units in position two and three are strongly dependent on the single unit occurring in the preceding position. This would suggest that linear order in chimpanzees would suffice to create longer sequences, notwithstanding transitional relationships between the internal elements.

Overall, the data suggest that chimpanzees might possess some fundamental combinatorial principles which can be used to establish sequential relationships across multiple sound units. The positional information in bigrams is relevant to the sequential ordering of the uttered message and most importantly, the first position seems to be filled with compulsory information when a certain unit needs to be introduced in the second position. This seems to be especially relevant for units like PS or PB, which occur less frequently in isolation and which are often preceded by the specific unit PH in bigrams.

The fact that chimpanzees have access to local ordering patterns and precedence relationships during spontaneous vocal production supports previous data coming from auditory sound discrimination paradigms in other primates, which showed sensitivity to ordering violations between adjacent sound units in artificial strings generated by simple finite-state grammars of the $(AB)^2$ type[11,14,80–82]. Beyond adjacency relationships,

chimpanzees and other primates have been shown to generalize over dependency rules between non-adjacent elements in both sound and visual discrimination tasks[13,83–85]. This suggests some ordering capacity that would go beyond simply adding two single units together in a sequence. In this study, we found that in trigrams, chimpanzees might also rely on positional information, where some over-represented bigrams within trigrams are consistently produced at the beginning of the sequence (first position, e.g., HO_PH + X, GR_PG + X, PH_PB + X). Similarly, a few over-represented bigrams—which occur above chance at both two-unit and three-unit level—appear to co-occur together with a third single unit, which generally linearly precedes them, when occurring in three-unit sequences (e.g., GR_PG + GR, HO + PH_PS, HO + PH_PB).

Beyond bigrams and trigrams, we also found 60 different sequences longer than three units, together representing 15.8% of all utterances emitted. There were not sufficient numbers of each to examine them in detail. Thus, the sizable sample of close to 5000 vocal utterances would likely need at least tripling to examine patterns in longer vocal sequences. With our large dataset, we can nevertheless be confident that the estimation of the frequency of production of sequences of different lengths is likely representative of the true vocal output even if sequence diversity beyond bigram or trigram is likely underrepresented. One interpretation of why sequences longer than three units are rarely produced might be that longer sequences are not important. Alternatively, one can argue that frequency does not necessarily equate to significance, as rare but crucial parts of vocal

repertoires are well documented. Alarm calls in social species, for example, are likely underrepresented in overall vocal emission compared to social spacing calls, such as contact calls[62].

Of note, two-unit and longer sequences emitted by both sexes occur throughout the chimpanzee vocal repertoire. They are not limited, for example, to loud calls, nor more specifically to the well-described four-unit pant-hoot sequence (Supplementary Fig. 4a–c, see supplementary discussion). Our approach thereby proves valuable in not only identifying the commonly described pant-hoot sequence but also sequences containing other units that follow nonrandom ordering patterns.

Although the current dataset is not small, determining the importance of these vocal sequences in chimpanzees' communicative needs will require even larger datasets. An important next step is to assess whether the production of a large diversity of sequences with ordering properties actually promotes versatile meaning generation. To this end, contexts of vocal production require detailed analysis. While some chimpanzee single units have demonstrated context-specificity in previous studies (e.g., Crockford[41]), the context-specificity of sequences remains largely uninvestigated (but see Leroux et al.[44]). In particular, whether the juxtaposition of single units into sequences leads to some meaning shifts is a promising area for future research. In fact, we identified several bigrams constitutive of the trigrams that were also produced as bigrams alone (i.e., without a third element, i.e., BK_SC, GR_PG, HO_PH, PH_PB, PH_PS, PS_SC, and WH_PH) whereas other bigrams within trigrams were never produced as bigram alone (i.e., PB_BK, PB_PR, PB_PS, PG_PB, PS_PB and SC_PS, Supplementary Fig. 10). Contextual information will therefore help understanding the kind of combinatorial system beyond the generative capacity in chimpanzees (e.g., Rizzi[86]): specifically, whether chimpanzees treat frequent bigrams as stored chunks to be reused in trigrams as a whole, or alternatively whether the transitional relationships between adjacent calls reflect simple adjunction or some more complex combinatorial mechanism beyond single combinations. Thus, a future goal is to determine whether the flexibility we have identified across the chimpanzee vocal repertoire relates to flexibility in the information conveyed. This will require analyses testing, first, for potential contextual and/or meaning shifts between single units emitted alone, and the same unit emitted paired with another single unit in a bigram. A second aim is to test whether the bigram produced alone undergoes a meaning shift when combined with third or fourth single units.

To assess the likelihood that other primate species may also have flexible vocal sequence capacities, we conducted a literature search. We found substantial evidence of vocal sequence production across all four primate clades, in both singing and nonsinging species (Fig. 6). For singing species, as in other taxa with monogamous breeding systems such as birds, monogamous primates demonstrate co-evolution of singing and duetting, likely due to sexual selection pressures. Across primate clades, singing usually occurs in territorial or courtship contexts[87,88], with vocal sequences containing 3–13 unit types (ref. [55] and Fig. 6). Addressing capacity (1), flexibility, units used in 'song' are rarely reported to be used singly, or in other parts of the vocal repertoire[28,29]. Addressing capacity (2), ordering, as in other singing taxa, such as birds and whales[89], it is thought that pattern variation of single units in the sequence primarily functions to identify individuals or groups and likely conveys little other socio-ecological contextual information[29,64,90].

For nonsinging primates (Fig. 6), vocal sequences production is observed across all four primate orders, suggesting it is a relatively common primate trait. However, sequences are usually short (2–5 single units) and mainly restricted to alarm calls (usually bigrams) or long-distance calls. Exceptions are olive colobus, marmosets, bonobos, and gorillas, interestingly all forest-dwelling species, where vocal sequences have been described in both intra-group and inter-group contexts[57,58,69,91]. This literature review demonstrates some use of vocal sequences across many primate species. Note that quantitative whole-repertoire analyses have rarely been conducted and may reveal broader use of structural complexity within species than we are currently aware of.

Nevertheless, as yet, only human and chimpanzee vocal systems seem to encompass all three of our structural capacities. This has implications for the meaning-generation potential of a species. Consider a vocal system with a vocal repertoire of six vocalizations and only one bigram. This gives the potential to encode seven different meanings (such as the chacma baboon[27]). The chimpanzee vocal system, consisting of 12 call types used flexibly as single units, or within bigrams, trigrams or longer sequences, offers the potential to encode hundreds of different meanings. Whilst this possibility is substantially less than the infinite number of different meanings that can be generated by human language, it nonetheless offers a structure that goes beyond that traditionally considered likely in primate systems. The versatile patterning of chimpanzee vocal sequences presents a valuable opportunity to examine whether detected patterns also relate to predictable meaning shifts. Further research challenges include comparing the contexts of production of single units (such as a bark), the single unit emitted paired with another single unit (bark + panted hoo), or a bigram emitted before or after a third single unit (hoo + [bark + panted hoo]; or [bark + panted hoo] + grunt).

Compared to compositionality in vocal sequences produced by animals, human language compositionality is based on hierarchical structure rather than linear order, where the structure is determined by the word categories being combined (e.g., nouns, verbs, prepositions forming noun phrases, verb phrases, or prepositional phrases, respectively). Whilst linear order can change meaning, and open up the potential for meaning generation in limited systems, a ubiquitous phenomenon in human language is indeed the fact that the same linear order may convey different meanings, depending on relevant kinds of the underlying structure. For example, the expression *the man drew a boy with a pencil* can either mean that a man used a pencil to draw a boy, or that he drew a boy who's holding a pencil. This is often taken as evidence that linear ordering alone is insufficient to capture language[18,92,93].

An influential hypothesis in linguistic theory states that the computational system holding hierarchical representations in human language—i.e., Merge—might be based on a very parsimonious computation, which builds together phrases and sentences from individual word categories and which is assumed/proposed to be neutrally hardwired in the human brain[94–96]. Some studies have raised the prospect of using Merge as way to describe animal vocal constructions at a higher degree of formalization[10,79]. In contrast to language, however, animal communication seems to lack any categorical dimension on the units of analysis as a hierarchical prerequisite (such as noun or verb equivalent)[93], although actual empirical tests of this have rarely be attempted. While cross-species comparability in the chimpanzee multiunit vocal sequences is premature, the positional bias, the transitional relationships, and the potential for recombination within a pairing system, such as emitting independently produced bigrams also within trigrams, points to the chimpanzee vocal sequences as a valuable system for analysis. Future studies must determine whether combining single units into bigrams and bigrams into longer sequences creates predictable contextual or meaning shifts.

**Conclusions**. Here, we reveal a highly versatile vocal system in a non-human animal, the chimpanzee, demonstrating flexible combination and recombination of single units across the vocal repertoire. Most calls in the vocal repertoire could be combined with most other calls so that single-use vocalizations were also emitted within bigrams, and independently produced bigrams were added to third units to produce trigrams. Although single units could occur in first or last positions, there were strong positional and transitional biases, consistent with clear ordering patterns. Previous studies report a highly limited capacity for animal vocal repertoires to produce flexible vocal sequences that can support numerous differentiated meanings. Our results examining chimpanzee vocal sequence structure suggest that these conclusions may be premature. Further research must show, however, to what extent these structural sequences signal predictable meanings.

## Methods

**Study site and subjects**. We conducted the study within the Taï Chimpanzee Project[97] on wild western chimpanzees at the Taï National Park, Côte d'Ivoire (5°45′N, 7°07′W). TB collected data on all adult and subadult chimpanzees from three communities (East, North, and South) between two study periods: January-February 2019 and December 2019 to March 2020. We defined as adult all chimpanzees ≥15 years of age and subadults as chimpanzees between 10 and 15 years of age. Adult and subadult chimpanzees are referred in our study as mature individuals. For this study TB collected data on 46 mature chimpanzees, 5 males and 10 females in East group, 4 males and 8 females in the North group, and 5 males and 14 females in the South group.

**Data collection**. TB followed the chimpanzees from dawn to dusk during c.a. 12 h per day. TB recorded vocalizations during half-day focal animal sampling[98] switching the focal animal around 12:30 pm, resulting in c.a. 6 h of continuous sampling per focal. Using a 2 s pre-record option, she audio recorded each vocalization from the focal chimpanzee as well as any vocalization produced by individuals visible around the focal animal for whom the identity of the caller could be identified with certainty ad libitum[98]. TB recorded the vocalizations using a Sennheiser ME67 directional microphone (digitized at a 48 kHz sampling rate and 24-bit sampling depth) connected to a Tascam DR-40X digital recorder. TB focalled mature chimpanzees for 513 h and collected ad libitum data for an additional 387.8 h. This resulted in mean ± SE 13.2 ± 0.9 SE hours of focal sampling on 39 mature focal individuals. In addition, vocal production from seven extra mature individuals was recorded ad libitum. Per individual, TB collected an overall mean ± SE of 34.4 ± 3.5 vocal utterances during focal hours and 85.89 ± 7.02 vocal utterances ad libitum. Per hour of focal data, TB obtained 3.9 ± 0.3 vocal utterances, from which 2.1 ± 0.2 where single units and 0.9 ± 0.1 were sequences (i.e., at least two different single units produced one after the other with less than 1-s pause between them). Of the single units, 1.8 ± 0.2 were non-panted single calls (HO; GR; BK; SC; WH; NV) and 0.4 ± 0.1 were panted single calls (PH; PG; PB; PS; PR).

**Construction of vocal repertoire**. The chimpanzee vocal repertoire consists of several call types. Most of these call types are emitted either singly or in a "panted" form whereby a voiced inhalation is inserted between each call (Supplementary Fig. 2a, b). Thus hoos can be emitted as single hoos, as repetitions of single hoos with 100–500 ms between each hoo, or as sequences of panted hoos, with c.a. 100 ms between each hoo and pant. Whilst each call type can be emitted singly, panted versions are only emitted as repetitions. Given that inserting pants between single calls often changes the context in which calls are emitted, we attribute panted versions as being different units. Thus, we divide the repertoire into call types and their panted versions resulting in seven single forms and five panted forms, here termed "single units" (see details in Table 1 and Supplementary Figs. 1 and 2). Each of these 12 single units can be combined, where different combinations result in different sequences. We defined a single unit as a unit type emitted alone or repeated within 2-s intervals. We defined a sequence as different types of single units emitted with less than 1 s interval (e.g., hoos followed by panted hoos and panted barks (Supplementary Fig. 3B); or panted hoos followed by panted barks, panted grunts and grunts (Supplementary Fig. 3C) emitted within 1 s). Different variants of the same call type (e.g., "rest" or "alert" hoos in Crockford et al.[46]) are not differentiated here but are considered as the same unit type.

**Assigning units to recorded vocal sequences**. Recordings are examined using PRAAT version 6.1.31 spectrograms which show the frequency distribution across the call[99]. An inherent problem in acoustic recordings in tropical forests is the dense background noise, making automated processes for extracting acoustic measurements problematic, especially for quieter calls such as hoos and grunts. However, different call types can be distinguished using spectrograms, which reveal both temporal and spectral properties[43,100]. Call types can be differentiated due to their distinctive acoustic features (Table 1, see SI for spectrograms and sound files of each call type, and gradations of these call types). For the analysis, we considered only calls of high quality, with the lowest frequency band visible, recorded from the beginning to the end, and with the signaller ID defined.

The chimpanzee vocal repertoire is a graded system (Supplementary Fig. 9), such that most call types grade into other call types. Call types that were thus difficult to categorize were sent to a blind coder and an expert in chimpanzee vocal repertoire (CC). If there was no agreement between at least two coders, the call was categorized as unclear. We did not include in the analysis utterances containing unclear calls. Utterances in which the start or end could not be coded due to overlap or recording omission were also not included. TB coded all the data. In total, 6% of the data (301 calls across all call types) was subjected to inter-ratter reliability with a blind coder. At the end of the training, TB and the blind coder reached a 94.6% of agreement on the call classification. In total, 4826 utterances were used for this study out of 5517 utterances recorded, comprising 401 differently constructed utterances which included one to ten different single units (see details in Supplementary Data 1). Our goal was to test if the three capacities laid out in the introduction apply to chimpanzees.

**Statistics and reproducibility**. Capacity 1—Flexible combination of single units within sequences: To test capacity 1, we extracted each vocal sequence from our dataset and quantified the diversity of these sequences, their length and which single units were used into sequences.

Capacity 2—Ordered production of single units within sequences: To test capacity 2, we assessed whether the sequences formed by two (bigrams) and three (trigrams) single units in our study were just a random juxtaposition of single units, or if they conversely resulted from some ordering rules/ nonrandom order among the individual calls. Sequences formed by three single units constituted the maximum vocal sequence length that was reached across all individuals (Fig. 1). Thus, the sample sizes of sequences longer than three units were overall too small to run meaningful statistical analysis on each respective length (i.e., four and then five and then six, etc.).

We asked three questions concerning the patterning organization of single units emitted in sequences of two single units (hereafter bigrams). (1) Positional bias: are the single units produced by chimpanzees biased towards a certain specific position within a bigram? (2) Specific ordering frequency: do the bigrams show fixed order frequencies that go beyond random juxtaposition of single units? (3) Transitional bias: do relationships between these units exist (i.e., υ follows ω, or υ precedes ω) when forming bigrams?

*Positional bias*. A unit υ occurring more often in position 1 than in position 2 (or vice versa), would be classified as having a potential positional bias towards position 1 (or vice versa). For each unit, we then conducted a Bayesian binomial test on JASP[101] with default effect size priors and flat Beta (1,1) to quantify the relative likelihood for a potential positional bias within the sequence. We defined the number of successes over all trials for each unit according to the number of occurrences found in the most frequent position. Results are reported for the Bayesian factor $BF_{10}$ testing the hypothesis that the proportion of occurrences in the most frequent position is higher than the default test value set at 0.5 (50%). Effect size estimates are reported as median posterior population ($\theta$) with credibility intervals (CI) set at 95%[102].

*Ordering frequency*. We used randomization routines to assess which bigrams were produced more than by chance. We first established the frequency of production of every single unit produced singly and in sequences (see Table 1). This constituted the frequency pool of observed frequency of call production. In this study, we recorded 817 bigrams (Fig. 1). We therefore sampled randomly from this this-tribution (i.e., the frequency pool) 817 pairs of single calls, with both calls being different to create 817 random bigrams. We then compared the distribution of the random bigrams to that of the observed bigrams (the bigrams that have been recorded), and identified which bigrams had an observed frequency above the random frequency. We repeated this process 1000 times to establish whether each bigram occurred more than by chance (i.e., more than by random juxtaposition of single units). Bigrams were considered to occur more than by chance if the observed frequency was above the randomized frequency in at least 950 randomizations (i.e., 95% of the randomizations).

*Transitional bias*. We took, for consistency, only the bigrams that were produced above-chance levels. We tested two different transitional relationships between the two single units υ and ω within the bigrams. First, we asked: given υ, how likely is it to find ω, compared to all other possible following units? For convenience, we called this transitional possibility a forward relationship. Second, we asked: given ω, how likely is it to find υ, compared to all possible preceding units? We called this alternative transitional possibility a backward relationship. We quantified the likelihood for a potential relationship bias between any two units using Bayesian binomial tests with the same parameters and results report as described in (1) above. The number of successes over all trials for each sequence was defined according to the number of times that υ was followed by ω (or that ω was preceded by υ), compared to the number of times that υ was followed by any other unit (or

that ω was preceded by any other unit). Since the test value for this Bayesian analysis was kept constant at 0.5, we measured how far above this level a certain call would either precede or follow another call, using a very conservative ratio of combinatorial patterning of 1:1 as a starting point. We acknowledge that each call in the repertoire could have been preceded or followed by more than two calls, so the ratio of each call to precede or follow another call is 1/total number of calls. We however reasoned that using a very conservative test value at 0.5 would have been more informative for two interdependent reasons: we would have been more confident to detect true transitional relationships in the chimpanzee's vocal system —which might have been masked with lower test values in a very flexible repertoire; we aimed at detecting a small set of highly consistent transitions as a starting point for future investigations—rather than offering a full-fledged description of the overall transitional patterns in the repertoire, which would require a larger sample. We thus investigated whether certain calls ω exist, which would either precede or follow υ at least 50% of the time, compared to all the other calls pooled together.

As a further test of capacity 2 beyond bigrams, and a test of capacity 3, we also assessed which three-unit sequences (trigrams) were produced above chance using a similar randomization procedure as for bigrams (see below). We analyzed which bigrams were reused into trigrams and if this occurred more than by chance as a test of capacity 3 (see below). Finally, to test capacity 2 further we also assessed if these bigrams were biased towards a certain position in the trigrams and if transitional bias existed between bigrams and single units within trigrams. For convenience, we detail all these analysis under the header criteria 3 below.

Capacity 3—Recombination of independently emitted sequences: To assess the recombination of independently emitted sequences we focused on how chimpanzees combine simple bigrams with a third unit to produce trigrams. We assessed which bigrams would appear more than by chance within trigrams. We first asked: (1) Specific ordering frequency: (a) Are the trigrams produced beyond simple random juxtaposition of single units? (b) Were some bigrams occurring within trigrams beyond simple random juxtaposition of single units? (2) Positional bias: are these bigrams biased towards a certain specific position (head position, tail position) within bigger trigrams? (3) Transitional bias: do relationships between these bigrams and single calls exist (i.e., υω follows δ, or υω precedes δ) when forming trigrams?

*Ordering frequency*. We repeated the same randomization process as described for bigrams above but for the trigrams. We extracted randomly 458 trigrams (i.e., the number of trigrams recorded for this study, Fig. 1) and compared this distribution to the distribution of observed trigrams. As for bigrams, we repeated this procedure 1000 times and the trigrams were considered to occur above-chance level if their observed frequency was above the randomized frequency in 950 randomizations. As for (1b) we extracted from each trigram the two possible bigrams within these sequences. For instance, a trigram HO_PH_PG would produce the two following bigrams: HO_PH, PH_PG (see Table 1 for the abbreviation of call names). We then assessed which of these bigrams in trigrams were produced more than by chance using the same approach as for the bigrams produced in sequences with only two single units. For this analysis, we sampled 916 bigrams 1000 times.

*Positional bias*. We restricted our examination to those bigrams which occurred above-chance levels in (1b) above. We counted the positional occurrence—head position, i.e., first and second position; tail position, i.e., second and third position —of each bigrams within each of the trigrams ($N = 406$). Each bigrams in a sequence was assigned either to head or tail position bias, according to its position of predominant occurrence. Thus, a trigram υωε that contains a bigram υω occurring more often in the head position than in the tail position, would be classified as having a potential positional bias towards the head position. For each sequence, we quantified the likelihood for a potential positional bias using Bayesian binomial tests with the same parameters and results report as described in capacity 2 above. The number of successes over all trials for each sequence was defined according to the number of bigrams found in the most frequent position.

*Transitional bias*. We again asked in one case if given υω, how likely it is to find ε, compared to all possible following units (forward relationship). In the other case, we asked: given υω, how likely is it to find ε, compared to all possible preceding units (backward relationship)? The potential relationship bias was assessed using frequentist and Bayesian binomial tests with the same parameters and results report as described in capacity 2 above. The number of successes over all trials for each utterance was thus defined according to the number of times that υω was followed by ε (or that υω was preceded by ε), compared to the number of times that υω was followed by any other units (or that υω was preceded by any other unit). To assess the bias, only the trigrams for which ε followed υω (or ε preceded υω) with a frequency higher than the sum of all other units following υω (or preceding υω) were included in the analysis.

We conducted all the randomization in R version 3.4.4[103] and the Bayesian analyses in JASP version 0.13.1[101].

**Literature review on sequence production in non-human primates**. Using Google Scholar, we searched peer-reviewed publications for information on vocal sequence production across primate taxa. We searched for each primate species by combining its scientific name and its vernacular name with the following keywords: vocal repertoire, vocal production, call, call combination, call sequence, vocal sequence, phonology, syntax, long call, vocal complexity. We identified vocal sequences either from the text or from accompanying spectrograms. We determined (1) whether sequences were reported for a given species, (2) if so what was the longest reported sequence and 3) what was the number of different sequences reported. The results of this search are depicted in Fig. 6 for species in which we could find any description for at least one sequence containing at least two different call types or notes (for singing species). We cannot exclude that we missed some publications that reported sequences in species not listed in Fig. 6. We also cannot exclude that the species indicated in Fig. 6 may emit longer vocal sequences or have a larger variety of sequences than is currently reported in the literature. For several species, it is not clear if the vocal units emitted in the sequences are also produced singularly or if they are only produced as part of a sequence. We did not consider repeats of the same vocal unit (e.g., A_A_A) as a sequence. For the sake of comparison with other species in which the methods differ slightly from our current approach, we recalculated for Fig. 6 the length and diversity of sequences excluding counts of repetitions of the same vocal unit at any time point within a sequence (e.g., we treated A_B_D as the same sequence as A_B_A_D). In this review, we differentiated singing from nonsinging species, as in singing species descriptions of vocal sequences tend to be limited to song contexts only, which are typically related only to sexual attraction or territorial contexts and may reveal little about the vocal sequence usage more generally.

**Ethic statement**. Our study was purely observational and non-invasive. Observers followed the strict hygiene protocol of Taï Chimpanzee Project, which was adopted by IUCN as the best practice guideline for wild ape studies[104] 3.44. Observers quarantined for 5 days before following the chimpanzees. During follows, observers disinfected their hands and boots and changed clothes before leaving and entering camps. In the forest, observers wore face masks and keep a minimum distance of eight meters between themselves and the chimpanzees, to avoid disease transmission from humans to chimpanzees, and to avoid disturbing the natural behavior of the observed individuals. The research presented here was approved by the 'Ethikrat' of the Max Planck Society on 04.08.2014.

**Reporting summary**. Further information on research design is available in the Nature Research Reporting Summary linked to this article.

## Data availability
The source data generated and/or analyzed during the current study are available from the corresponding author on reasonable request. The data used to generate the figures in this manuscript are provided in Supplementary Data 2.

## Code availability
No custom computer code or algorithm was generated to analyze the data in this article. The R and JASP codes for the analysis are available from the corresponding author on reasonable request.

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

## Acknowledgements

We thank the Ministère de l'Enseignement Supérieur et de la Recherche Scientifique and the Ministère de Eaux et Fôrets in Côte d'Ivoire, and the Office Ivoirien des Parcs et Réserves for permitting the study. We are grateful to the Centre Suisse de Recherches Scientifiques en Côte d'Ivoire for their logistical support, and to Kayla Kolff and the staff members of the Taï Chimpanzee Project for their support and assistance in collecting the data, and to Christophe Boesch for tremendous work in establishing and nurturing the Taï Chimpanzee Project for 30 years. This study was funded by the Max Planck Society and the European Research Council (ERC) under the European Union's Horizon 2020 research and innovation program awarded to C.C. (grant agreement no. 679787). Core funding for the Taï Chimpanzee Project was provided by the Swiss National Foundation (1979–1997) and Max Planck Society (since 1997). We thank Julia Fischer, Cat Hobaiter and Adriano Lameira for comments extremely helpful in shaping this manuscript.

## Author contributions

A.D.F., C.C., C.G.B., E.Z., and R.M.W. designed the study; T.B. collected the data; C.G.B. and E.Z. conducted the statistical analyses; C.C., C.G.B., and E.Z. wrote the first draft of the manuscript with substantial input from all co-authors.

## Funding

## Competing interests

The authors declare no competing interests.
