## [Peer Review File · Communications Biology]

Reviewers' comments:

Reviewer #1 (Remarks to the Author):

The manuscript presents an interesting and thorough quantitative analysis of the vocal sequences produced by chimpanzees in the Tai National Park. The authors report that most of the individual vocalizations made by these animals are combined into two, three or even longer call utterances. They also show that these combinations are not random, with some bigrams (pairs of calls) occurring more frequently than others, and that these bigrams are again non-randomly combined into longer sequences. I find this interesting, and a nice addition to the literature.

However, I do have concerns about how this paper is presented. The abstract and introduction begin with explaining how the ability to flexibly combine phonemes into words and phrases gives rise to the unlimited expressivity of human language. They then describe what appears to be related type of non-random vocal combinatoriality in chimpanzees. However, the reason that the combinatorial system of human language is so powerful is because the units which are being combined carry distinct meanings, combined to create novel meaningful utterances. The authors do note this in places, but in other areas of the manuscript I think it would very easy for a reader to overlook this very substantial difference between human and chimpanzee communication, and that the text should be revised to make this clear.

It is true that if humans lacked this combinatorial system, they would have a greatly reduced communicative capacity. However it is also true that it does not matter how combinatorial a system may be, if the component calls (or call combinations) do not have meaning, and the animals do not have the ability to flexibly (and mutually) attribute new meanings to new combinations of calls, this system offers no meaning generation potential.

While nonhuman primates do produce certain calls in certain context (including alarm calls, food calls, and calls used in various social contexts), I am aware of no evidence of primates flexibly attaching new meanings to their vocalizations.

If a new object is introduced into their environment, chimpanzees do not assign a new call sequence to refer to that object, and use this in their communication with one another. Evidence of this behavior would represent a huge advance in animal communication.

I know that the authors are aware of this, and that they make some concessions to this point in the manuscript (phrases like 'From a purely structural perspective...', 'An important evolutionary step towards language', '...meaning generation potential', etc.).

However, the manuscript also makes statements like the following:

'Unique to human language is its capacity to combine a limited sound set into words, and then flexibly and hierarchically structure words into phrases, allowing the generation of endlessly new sentences and thereby new meanings...However, unlike humans, animal capacities to generate vocal sequences (hereafter "sequences") appear highly constrained where a sequence is broadly defined as the production of two or more different types of single vocal units within a short time of each other'

'...Chimpanzees might form combined meanings from two individual units, which can, in turn, be recombined with a third unit to eventually output a third combined meaning'

I do not think it would be unreasonable for a reader to interpret these statements as meaning that if animals were able to generate more complex vocal sequences, they too might be able to generate endless new meanings.

To avoid such confusions and possible misinterpretations of these data, the authors should revise the text and add a section discussing the difference between human language and chimpanzee

communication in terms of the types of meanings that can be conveyed. Without this, a huge part of the story on the evolution of human language (form-meaning mapping, which I would argue is a much larger piece of the evolutionary puzzle than the ability to non-randomly combine calls), is missing from this manuscript.

Minor comments:

1. The authors note that humans and other animals have vocal sound sets of overlapping size. (English has about 42 phonemes, the chimpanzees reported here have 12.) However, they also note that certain calls are produced in certain contexts, which seems to suggest another difference between phonemes, which do not typically carry meaning in isolation. I wonder if the fact that at least some chimpanzee calls are used in certain contexts would limit their ability to flexibly use them in novel contexts?

2. (lines 89-90) '...if differing ordering of the same single units should encode different information...'

I think this is one of the critical points, does the order and composition of the calls reported here affect meaning? I understand that answering this question is outside of the scope of the current manuscript, but the authors could at least make clear that this is very much an open question.

3. Methods: I understand the decision to use a baseline of 0.5 (50%) for some analyses. For example, in the analysis of positional bias, there are only two positions in a bigram in which a call can appear. However, it appears that the same baseline is used in the transitional probability analyses, apparently because it is the default value(?). If one call transitions to another more than 50% of the time, that does seem like an important result. But if anything, this seems quite conservative, given that each vocalization could be followed by ~10 other different calls, I am not sure how the value of 50% was calculated. A little more clarity on how these values were selected would be useful.

Reviewer #2 (Remarks to the Author):

This paper performed the large-scale quantitative analyses of vocal sequences produced by chimpanzees. Specifically, the authors first defined three structural criteria and, correspondingly, evaluated those criteria against chimpanzee vocal sequences, in addition to the interesting meta-analysis of the literature. The paper is well-organized and well-written, so I would recommend publication of this paper in *Communications Biology*, provided that the following major and minor comments were sufficiently addressed.

Major comments:

p.24: One possibility remains that once chimpanzees produce (or process) particular bigrams many times, those bigrams are stored and reused as a chunk, where generative capacity of chimpanzees is limited to only one merger at a time. For example, once A and B are merged into AB many times, AB is stored as a chunk, which can in turn be merged with C into ABC, and ABC will be stored as a chunk, and so on. In other words, A, B, and C cannot be merged at the same time, so in order to create ABC, the bigrams like AB or BC are required in advance. Then, this possibility makes the interesting testable prediction that there must exist constituent n-grams in order to create n+1-grams, and I wonder whether this is actually the case. Specifically, if trigrams like ABC exist, then there must be bigrams like AB or BC. And in the same vein, if quadgrams like ABCD exist, then there must be trigrams like ABC or BCD. Otherwise, chimpanzees should have produced bigrams/trigrams first in mental workspace to create trigrams/quadgrams, leading to the stronger conclusion on generative capacity of chimpanzees.

p.27: Whether three structural criteria defined at the beginning of the paper are actually observed in human language is not clear in the first place from the quantitative perspective. In order to address this question, combinatorial networks, positional and transitional biases, etc. should be

computed based on text data of human language and quantitatively compared with those of chimpanzees in order to investigate whether combinatorial properties of chimpanzees are human-like or not.

Minor comments:

- I.129: but see 26 -> fix superscripts
- I.131: but see 26,27 -> fix superscripts
- I.137: criteria three -> the third criteria
- I.228: 10 units -> ten units
- I.513: is -> are
- I.518: 390 unique sequences -> not discussed in Results? (only in Abstract)
- I.536: was -> were
- I.537: Third we -> Third, we
- I.541: explain "biologically-induced auto-correlation effects"
- I.588: GR_PG + GR -> GR + PG_GR?
- I.604: e.g., 43 -> fix superscripts
- I.612: no connectives between sentences
- I.631: function -> functions
- I.643: fix the structure of the sentence
- I.671: produce -> produced
- I.683: remove two commas
- I.684: show however to -> show, however, to
- I.685: meanings -> meanings.

Reviewer #3 (Remarks to the Author):

This paper investigates the structural complexity of chimpanzee vocal sequences. Results show that chimpanzees produce a wide variety of vocal sequences in their natural communication and recombine them across their vocal repertoire.

The strong point of this paper is that the authors use the vast data set of vocal production and analyzed their combinatorial structure using a variety of methods. The weak point is that this study is descriptive and lacks either analyses of context dependency of call production or playback experiments. In order to explore evolutionary continuity between chimpanzee vocal sequences and human hierarchical syntax, it is necessary to explore whether call combinations produce a compositional message to receiver animals (see Suzuki et al. 2019 Phil Tran R Soc B). Indeed, although hierarchical structures of vocal sequences have been demonstrated for songs of many passerine species (e.g., Sainburg et al. 2019 Nat Comm) and whales (Cholewiak et al. 2013 Mar Mammal Sci), these sounds seem not provide compositional information, but could be categorized as phonology (combinations of meaningless sounds). Or, even if the combinations of calls are context dependent, receivers may recognize it as an idiomatic sequences with a new, third meaning (Arnold & Zuberbuhler 2012 Brain Lang). Thus, simply examining structural complexity is not enough to claim that the observed sequences are parallel with compositionality or hierarchical syntax. I think the authors should discuss the possibility that observed vocal sequences in chimpanzees are phonology or idiomatic sequences, rather than compositional expressions.

Regardless of the lack of analyses from the receiver side, this paper is enjoyable to read and provides important data on vocal combinations in our closely related species. Minor comments are below:

Line 125: "Japanese great tits" have recently been renamed as "Japanese tits".

Line 217ff: Have you tried to conduct discriminate function analyses to distinguish between different vocal elements?

Line 653ff: The definition of "compositionality" does not require hierarchical processing. It is defined as the process in which "the meaning of a combinatorial expression is determined by the meanings of its constituent parts and the rules used to combine them".

Reviewer #4 (Remarks to the Author):

I think this paper by Girard-Buttoz et al. represents an important contribution to the emerging field of animal combinatorics through quantifying the combinatorial dynamics of the chimpanzee vocal system. I found the paper to be well written and the justification for the study to be generally sound.

Aside from a few smaller issues regarding, for example, discussion of previous work (see below), my main concern is that I am not convinced that the way the authors have carved up the repertoire is fully consistent with current understanding of chimpanzee vocal behaviour. I am unaware of any studies where pant variants of vocalisations have been considered as stand-alone call entries in the vocal repertoire of chimpanzees (other than the pant-grunt), complicating subsequent analyses of the combinatorial dynamics of those single units. To assess how problematic this is, I think it would be important for the authors to re-run the analyses without these pant variants and with an inventory of calls that more accurately reflects the chimpanzee repertoire and assess the extent to which the broad patterns the authors argue exist then hold.

Minor points:

L68-69: What about song repertoire research in birds and cetaceans? Have these studies not also quantified sequence structure across the repertoire?

L72: I'm not sure this is accurate. There is compelling research in whale song and bird song that the structures cannot be captured by simple Markovian dynamics and require non-Markovian (i.e. hierarchical-like) models to describe the variation (see Suzuki et al. 2006 and Sainburg et al. 2019).

L84: What constitutes "most" here?

L142-143: This sentence seems to jump out of nowhere and reads a little tagged on. Might be better to integrate this point more into the discussion?

We are very thankful to the reviewers for providing very detailed and constructive comments that provide us to greatly improve the manuscript. Please find below the reviewers' comments in italic and our reply to the comments in bold.

Reviewer #1 (Remarks to the Author):

The manuscript presents an interesting and thorough quantitative analysis of the vocal sequences produced by chimpanzees in the Tai National Park. The authors report that most of the individual vocalizations made by these animals are combined into two, three or even longer call utterances. They also show that these combinations are not random, with some bigrams (pairs of calls) occurring more frequently than others, and that these bigrams are again non-randomly combined into longer sequences. I find this interesting, and a nice addition to the literature.

Thank you for this positive evaluation of our manuscript.

However, I do have concerns about how this paper is presented. The abstract and introduction begin with explaining how the ability to flexibly combine phonemes into words and phrases gives rise to the unlimited expressivity of human language. They then describe what appears to be related type of non-random vocal combinatoriality in chimpanzees. However, the reason that the combinatorial system of human language is so powerful is because the units which are being combined carry distinct meanings, combined to create novel meaningful utterances. The authors do note this in places, but in other areas of the manuscript I think it would very easy for a reader to overlook this very substantial difference between human and chimpanzee communication, and that the text should be revised to make this clear.

We very much agree that capacity to encode many meanings in a vocal system obviously requires both the ability to link sounds to meaning AND to have a suitably complex structure in which to embedded such meaning-bearing sounds. We had hoped that we had been clear about this as well as about the reasons why in this paper we focus only on the structure – and not on meaning. It seems that we have not entirely managed this and have worked to address this problem throughout the manuscript, including adding a final sentence to the abstract: *'Further research must show to what extent these structural sequences signal predictable meanings'* (Lines 44-45).

Other additions are:

***"While both structure and meaning are crucial requirements for flexible meaning generation, a way to establish a link between the production and sequence perception findings discussed above, is to assess whether the structure of vocal sequences across a vocal repertoire would facilitate the embedding of meaning, should meaning content be evident in the vocalisations."* Line 94-97.**

“We do not assess meaning in this study, but it is important to note that chimpanzee single unit use can show high context-specificity across a relatively broad range of contexts compared to other species”. Lines 166-168.

“If these single units carry meaning, it may be that chimpanzees form combined meanings from two single units, which can, in turn, be recombined with a third unit to eventually output a third combined meaning”. Lines 603-605.

“Future studies must determine whether combining single units into bigrams and bigrams into longer sequences creates predictable contextual or meaning shifts”. Lines 713-714.

It is true that if humans lacked this combinatorial system, they would have a greatly reduced communicative capacity. However it is also true that it does not matter how combinatorial a system may be, if the component calls (or call combinations) do not have meaning, and the animals do not have the ability to flexibly (and mutually) attribute new meanings to new combinations of calls, this system offers no meaning generation potential.

As mentioned above, we absolutely agree with you, and to make this clearer we now add into the first paragraph:

“Rather, the animal capacity to generate communicative sequences is considered highly constrained, both in terms of structural systematicity and meaning generation¹. In human language, words together with syntactic hierarchical structures permit flexible meaning construction. Such hierarchical structures have not been demonstrated in other animal communication systems although some animal calls show a limited capacity to encode meaning, and meaning can shift when single calls are emitted in short sequences²”. Lines 65-70.

While nonhuman primates do produce certain calls in certain context (including alarm calls, food calls, and calls used in various social contexts), I am aware of no evidence of primates flexibly attaching new meanings to their vocalizations.

Whilst it is true that there are few examples of primates producing totally novel vocalizations, there is growing evidence that they can refine meaning of a call. This is particularly evident with alarm calls – and makes evolutionary sense when different populations of the same species may face different predators. This was recently beautifully demonstrated in a drone experiment with green monkeys, showing that green monkeys, who had never seen a drone before, used an aerial predator alarm call upon seeing a drone³. Also we have previously noted in chimpanzees that alert calls to snakes are also emitted to snares⁴ (which are presumably a relatively recent threat to add to their repertoire of threats).

If a new object is introduced into their environment, chimpanzees do not assign a new call sequence to refer to that object, and use this in their communication with one another. Evidence of this behavior would represent a huge advance in animal communication.

It is true, we do not know of examples where such calls are also integrated into vocal sequences. However, there is an example of a zoo chimpanzee that began adding a raspberry (loud but voiceless blowing between compressed lips) to a pant hoot sequence, which other chimpanzees in the enclosure then also began to produce⁵. This pant hoot variant has not been heard in decades of research in at least two wild chimpanzee populations. Thus, it seems that chimpanzees may have some capacity to rearrange their limited and fixed set of call types into different sequences through the process of vocal usage learning.

I know that the authors are aware of this, and that they make some concessions to this point in the manuscript (phrases like ‘From a purely structural perspective...’, ‘An important evolutionary step towards language’, ‘...meaning generation potential’, etc.).

However, the manuscript also makes statements like the following:

We have rephrased the mentioned sections of the manuscript as follow:

‘Unique to human language is its capacity to combine a limited sound set into words, and then flexibly and hierarchically structure words into phrases, allowing the generation of endlessly new sentences and thereby new meanings...However, unlike humans, animal capacities to generate vocal sequences (hereafter “sequences”) appear highly constrained where a sequence is broadly defined as the production of two or more different types of single vocal units within a short time of each other’

Rephrased as: “Unique to human language is its capacity to combine a limited sound set into words, and to combine words into rule based hierarchically structured phrases, allowing the generation of endlessly new sentences and thereby new meanings..... Rather, the animal capacity to generate communicative sequences is considered highly constrained, both in terms of structural systematicity and meaning generation¹. In human language, words together with syntactic hierarchical structures permit flexible meaning construction. Such hierarchical structures have not been demonstrated in other animal communication systems although some animal calls show a limited capacity to encode meaning, and meaning can shift when single calls are emitted in short sequences². (Lines 60-70).

‘...Chimpanzees might form combined meanings from two individual units, which can, in turn, be recombined with a third unit to eventually output a third combined meaning’

Rephrased as: “If these single units carry meaning, it may be that chimpanzees form combined meanings from two single units, which can, in turn, be recombined with a third unit to eventually output a third combined meaning”. (Lines 603-605).

I do not think it would be unreasonable for a reader to interpret these statements as meaning that if animals were able to generate more complex vocal sequences, they too might be able to generate endless new meanings.

To avoid such confusions and possible misinterpretations of these data, the authors should revise the text and add a section discussing the difference between human language and chimpanzee communication in terms of the types of meanings that can be conveyed. Without this, a huge part of the story on the evolution of human language (form-meaning mapping, which I would argue is a much larger piece of the evolutionary puzzle than the ability to non-randomly combine calls), is missing from this manuscript.

Minor comments:

1. The authors note that humans and other animals have vocal sound sets of overlapping size. (English has about 42 phonemes, the chimpanzees reported here have 12.) However, they also note that certain calls are produced in certain contexts, which seems to suggest another difference between phonemes, which do not typically carry meaning in isolation. I wonder if the fact that at least some chimpanzee calls are used in certain contexts would limit their ability to flexibly use them in novel contexts?

This is indeed a very interesting point, that humans have a dual combinatorial system (combining phonemes into words, and then words into sentences). It is unclear whether animal calls operate synonymously with either one or the other of these options. Given that single calls can be context-specific, and yet combinations may either keep⁶ or change their meaning⁷, it may be too early in animal communication research to determine whether calls act more like words or can also act like phonemes, building words. Given this uncertainty, we remain agnostic on this point. Nonetheless, our new addition in paragraph 1 in the introduction, hopefully makes the position clearer, that whether animal sounds act as phonemes or words, their ability to generate flexible meanings is predicated on the capacity to combine sounds into sequences. We write: “Thus, it is unlikely that, for animals, the size of the sound set is the factor limiting meaning generation. Rather, the animal capacity to generate communicative sequences is considered highly constrained, both in terms of structural systematicity and meaning generation^{1”}. Lines 64-67.

2. (lines 89-90) ‘...if differing ordering of the same single units should encode different information...’ I think this is one of the critical points, does the order and composition of the calls reported here affect

meaning? I understand that answering this question is outside of the scope of the current manuscript, but the authors could at least make clear that this is very much an open question.

We agree with the reviewer that this remains an open question and we specifically added a sentence in the abstract and in the discussion to specify that: “Future studies must determine whether combining single units into bigrams and bigrams into longer sequences creates predictable contextual or meaning shifts.” Lines 713-714.

3. Methods: I understand the decision to use a baseline of 0.5 (50%) for some analyses. For example, in the analysis of positional bias, there are only two positions in a bigram in which a call can appear. However, it appears that the same baseline is used in the transitional probability analyses, apparently because it is the default value(?). If one call transitions to another more than 50% of the time, that does seem like an important result. But if anything, this seems quite conservative, given that each vocalization could be followed by ~10 other different calls, I am not sure how the value of 50% was calculated. A little more clarity on how these values were selected would be useful.

We have now clarified our decision criteria in the text of the method section as follow:

“We acknowledge that each call in the repertoire could have been preceded or followed by more than two calls, so the ratio of each call to precede or follow another call is 1/total number of calls. We however reasoned that using a very conservative test value at 0.5 would have been more informative for two interdependent reasons: we would have been more confident to detect true transitional relationships in the chimpanzee’s vocal system—which might have been masked with lower test values in a very flexible repertoire; we aimed at detecting a small set of highly consistent transitions as a starting point for future investigations—rather than offering a full-fledged description of the overall transitional patterns in the repertoire, which would require a larger sample. We thus investigated whether certain calls ω exist, which would either precede or follow υ at least 50% of the times, compared to all the other calls pooled together.” Lines 833-842.

Reviewer #2 (Remarks to the Author):

This paper performed the large-scale quantitative analyses of vocal sequences produced by chimpanzees. Specifically, the authors first defined three structural criteria and, correspondingly, evaluated those criteria against chimpanzee vocal sequences, in addition to the interesting meta-analysis of the literature. The paper is well-organized and well-written, so I would recommend publication of this paper in Communications Biology, provided that the following major and minor comments were sufficiently addressed.

We thank the reviewer for this positive evaluation of the manuscript.

Major comments:

p.24: One possibility remains that once chimpanzees produce (or process) particular bigrams many times, those bigrams are stored and reused as a chunk, where generative capacity of chimpanzees is limited to only one merger at a time. For example, once A and B are merged into AB many times, AB is stored as a chunk, which can in turn be merged with C into ABC, and ABC will be stored as a chunk, and so on. In other words, A, B, and C cannot be merged at the same time, so in order to create ABC, the bigrams like AB or BC are required in advance. Then, this possibility makes the interesting testable prediction that there must exist constituent n-grams in order to create n+1-grams, and I wonder whether this is actually the case. Specifically, if trigrams like ABC exist, then there must be bigrams like AB or BC. And in the same vein, if quadgrams like ABCD exist, then there must be trigrams like ABC or BCD. Otherwise, chimpanzees should have produced bigrams/trigrams first in mental workspace to create trigrams/quadgrams, leading to the stronger conclusion on generative capacity of chimpanzees.

We very much like this suggestion. In this paper, we have however limited our analyses to bigrams and trigrams. The reason is that although we used a corpus of nearly 5000 utterances, the sample size of sequences with four or more units was relatively small and would make detailed analyses of bigram and trigram use within these less instructive. We hope to be able to collect more data in a second work phase in the future.

p.27: Whether three structural criteria defined at the beginning of the paper are actually observed in human language is not clear in the first place from the quantitative perspective. In order to address this question, combinatorial networks, positional and transitional biases, etc. should be computed based on text data of human language and quantitatively compared with those of chimpanzees in order to investigate whether combinatorial properties of chimpanzees are human-like or not.

This is a good point - we have now added references from the human literature justifying each of our three structural criteria: “Using a comparative perspective, we examine these three lower-level but universal capacities in human speech. These capacities loosely reflect a system that develops in early childhood as a pathway to hierarchical syntax: corpora studies show how words, initially produced in isolation, are flexibly assembled into two-word phrases, linearly ordered in the language of use, and then recombined into longer sequences^{8,9}” (Lines 131-135).

.

Minor comments:

All the comments below were addressed and the correction made.

l.129: but see 26 -> fix superscripts
l.131: but see 26,27 -> fix superscripts
l.137: criteria three -> the third criteria
l.228: 10 units -> ten units
l.513: is -> are
l.518: 390 unique sequences -> not discussed in Results? (only in Abstract)

This results has been added to the result section.

l.536: was -> were
l.537: Third we -> Third, we
l.541: explain "biologically-induced auto-correlation effects"
l.588: GR_PG + GR -> GR + PG_GR?
l.604: e.g., 43 -> fix superscripts
l.612: no connectives between sentences
l.631: function -> functions
l.643: fix the structure of the sentence
l.671: produce -> produced
l.683: remove two commas
l.684: show however to -> show, however, to
l.685: meanings -> meanings.

Reviewer #3 (Remarks to the Author):

This paper investigates the structural complexity of chimpanzee vocal sequences. Results show that chimpanzees produce a wide variety of vocal sequences in their natural communication and recombine them across their vocal repertoire.

The strong point of this paper is that the authors use the vast data set of vocal production and analyzed their combinatorial structure using a variety of methods.

We thank the reviewer for this positive evaluation.

The weak point is that this study is descriptive and lacks either analyses of context dependency of call production or playback experiments. In order to explore evolutionary continuity between chimpanzee vocal sequences and human hierarchical syntax, it is necessary to explore whether call combinations produce a compositional message to receiver animals (see Suzuki et al. 2019 Phil Tran R Soc B). Indeed, although hierarchical structures of vocal sequences have been demonstrated for songs of many passerine species (e.g., Sainburg et al. 2019 Nat Comm) and whales (Cholewiak et al. 2013 Mar Mammal Sci), these sounds seem not provide compositional information, but could be categorized as phonology (combinations of meaningless sounds). Or, even if the combinations of calls are context dependent,

receivers may recognize it as an idiomatic sequences with a new, third meaning (Arnold & Zuberbuhler 2012 Brain Lang). Thus, simply examining structural complexity is not enough to claim that the observed sequences are parallel with compositionality or hierarchical syntax. I think the authors should discuss the possibility that observed vocal sequences in chimpanzees are phonology or idiomatic sequences, rather than compositional expressions.

Regardless of the lack of analyses from the receiver side, this paper is enjoyable to read and provides important data on vocal combinations in our closely related species. Minor comments are below:

We absolutely agree, and also do not make claims that we observe either compositionality or any form of syntax. We have now tried to make this clearer throughout. Our premise is simple: to generate diverse meanings when using a limited vocal repertoire, a complex structure is required in which meaning-bearing sounds/words/calls can be embedded. Here we simply examine whether there is any complexity in the vocal sequencing, in terms of flexibility, ordering and recombination. We make no claims at any point related to meaning nor to syntax. I hope this is clearer now.

Likewise, we make no claims as to how this structure is suggestive of phonological, idiomatic or compositional utterances. It is obviously not possible to make such claims until the contexts of production of the vocal sequences have also been investigated. We try to make this point clearer now also.

Line 125: "Japanese great tits" have recently been renamed as "Japanese tits".

We have changed it to Japanese tits throughout the manuscript.

Line 217ff: Have you tried to conduct discriminate function analyses to distinguish between different vocal elements?

Yes indeed Grawunder et al. 2021¹⁰ is now published and cited in the text (Line 210). Using both cluster analyses and discriminant function analyses, there is good discrimination of call types.

Line 653ff: The definition of "compositionality" does not require hierarchical processing. It is defined as the process in which "the meaning of a combinatorial expression is determined by the meanings of its constituent parts and the rules used to combine them".

We did not mean to infer that compositionality requires hierarchical processing; we have rephrased the sentence to clarify this point.

"Compared to compositionality in vocal sequences produced by animals, human language compositionality is based on hierarchical structure rather than linear order, where the structure is

determined by the word categories being combined (e.g., nouns, verbs, prepositions forming noun phrases, verb phrases, or prepositional phrases, respectively).” Lines 693-696.

Reviewer #4 (Remarks to the Author):

I think this paper by Girard-Buttoz et al. represents an important contribution to the emerging field of animal combinatorics through quantifying the combinatorial dynamics of the chimpanzee vocal system. I found the paper to be well written and the justification for the study to be generally sound.

We thank the reviewer for this positive evaluation of our manuscript.

Aside from a few smaller issues regarding, for example, discussion of previous work (see below), my main concern is that I am not convinced that the way the authors have carved up the repertoire is fully consistent with current understanding of chimpanzee vocal behaviour. I am unaware of any studies where pant variants of vocalisations have been considered as stand-alone call entries in the vocal repertoire of chimpanzees (other than the pant-grunt), complicating subsequent analyses of the combinatorial dynamics of those single units. To assess how problematic this is, I think it would be important for the authors to re-run the analyses without these pant variants and with an inventory of calls that more accurately reflects the chimpanzee repertoire and assess the extent to which the broad patterns the authors argue exist then hold.

We understand the confusion regarding the use of panted calls as stand-alone vocal units. We have now expanded the section in the manuscript motivating our choice: “Our rationale for treating panted-calls as separate call types rather than as a sequence of e.g. grunt and pant, was motivated by previous studies demonstrating that pant-grunt and pant-hoot are clearly stand-alone call types^{11,12} (reviewed in Crockford 2019¹³). For consistency in the treatment of panted call types, we include panted barks and panted screams as stand-alone call types. Please note that panted screams and panted barks are also reported across chimpanzee populations¹³.” Lines 218-223.

Also, from an analytical point of view, the suggestion of the reviewer would require to code panted calls as long sequences in themselves, because the pant and grunt components always alternate e.g. grunt_pant_grunt_pant_grunt_pant_grunt etc. In our view, this would create artificially long sequences that may over-represent the diversity of sequences in the repertoire. In addition, grunt_pant_grunt is functionally not different from grunt_pant_grunt_pant_grunt_pant_grunt because they are both pant grunt vocalizations addressed in a submissive context towards higher ranking individuals. However, following the reviewer’s suggestion they would be treated as two separated sequences, which is not desirable.

In sum, our approach is conservative and if anything under-evaluates the vocal sequence repertoire size of chimpanzees. We would therefore like to keep the analysis as is for the reasons exposed above.

Minor points:

L68-69: What about song repertoire research in birds and cetaceans? Have these studies not also quantified sequence structure across the repertoire?

Yes, this is certainly true of singing species – however meaning content is usually not attached to units of song, and hence would not be a relevant example here. We have clarified that this has been studied in singing species but we focus here on non-singing species where there are some examples of combining meaning bearing units or call types (see reply to the comment below as well).

L72: I'm not sure this is accurate. There is compelling research in whale song and bird song that the structures cannot be captured by simple Markovian dynamics and require non-Markovian (i.e. hierarchical-like) models to describe the variation (see Suzuki et al. 2006 and Sainburg et al. 2019).

Yes, this is certainly true of singing species – however meaning content is usually not attached to units of song, and hence would not be a relevant example here. To clarify this, we have rephrased the start of the paragraph to emphasize that we focus on non-singing species where there are some examples of combining meaning bearing units (unlike singing birds and whales which combine meaningless notes or syllables). We incorporated the citation suggested by the reviewer.

L84: What constitutes “most” here?

We have rephrased it here to specify that we mean the “vast majority”.

L142-143: This sentence seems to jump out of nowhere and reads a little tagged on. Might be better to integrate this point more into the discussion?

We have removed this sentence.

Bibliography

1. Townsend, S. W., Engesser, S., Stoll, S., Zuberbühler, K. & Bickel, B. Compositionality in animals and humans. *PLoS Biol.* **16**, 1–7 (2018).

2. Engesser, S. & Townsend, S. W. Combinatoriality in the vocal systems of nonhuman animals. *WIREs Cogn. Sci.* **10**, e1493 (2019).
3. Wegdell, F., Hammerschmidt, K. & Fischer, J. Conserved alarm calls but rapid auditory learning in monkey responses to novel flying objects. *Nat. Ecol. Evol.* **3**, 1039–1042 (2019).
4. Crockford, C., Wittig, R. M., Mundry, R. & Zuberbuehler, K. Wild chimpanzees inform ignorant group members of danger. *Curr. Biol.* **22**, 142–146 (2012).
5. Marshall, A. J., Wrangham, R. W. & Arcadi, A. C. Does learning affect the structure of vocalizations in chimpanzees? *Anim. Behav.* **58**, 825–830 (1999).
6. Leroux, M. *et al.* Chimpanzees combine pant hoots with food calls into larger structures. *Anim. Behav.* **179**, 41–50 (2021).
7. Ouattara, K., Lemasson, A. & Zuberbühler, K. Campbell's monkeys concatenate vocalizations into context-specific call sequences. *Proc. Natl. Acad. Sci.* **106**, 22026–22031 (2009).
8. Yang, C. Ontogeny and phylogeny of language. *Proc. Natl. Acad. Sci.* **110**, 6324–6327 (2013).
9. Yang, C., Crain, S., Berwick, R. C., Chomsky, N. & Bolhuis, J. J. The growth of language: Universal Grammar, experience, and principles of computation. *Neurosci. Biobehav. Rev.* **81**, 103–119 (2017).
10. Grawunder, S. *et al.* Chimpanzee vowel-like sounds and voice quality suggest formant space expansion through the hominoid lineage. *Philos. Trans. R. Soc. B Biol. Sci.* **377**, 20200455 (2022).
11. Goodall, J. *The Chimpanzees of Gombe: Patterns of Behavior*. (Harvard University Press, 1986).
12. Notman, H. & Rendall, D. Contextual variation in chimpanzee pant hoots and its implications for referential communication. *Anim. Behav.* **70**, 177–190 (2005).
13. Crockford, C. Why Does the chimpanzee vocal repertoire remain poorly understood? - and what can be done about it. in *The chimpanzees of the tai forest: 40 years of research* (eds. Boesch, Christophe, R. M. *et al.*) 394–409 (Cambridge University Press, 2019).

Reviewers' comments:

Reviewer #1 (Remarks to the Author):

The authors have done a good job in addressing my comments in this revision of the manuscript. My critical point, that this analysis does not consider the meaning of the calls or call combinations, is now much clearer. I am happy to recommend publishing this paper, which I am sure will be well received by the community

Reviewer #2 (Remarks to the Author):

I thank the authors for addressing the comments.

The second of the two major comments in the first review was sufficiently addressed, by referring to the literature on human language acquisition. However, the first of the two major comments was not addressed, unfortunately. I perfectly understand that n-grams beyond trigrams cannot be analyzed in the paper due to data sparsity, but the alternative possibility suggested in the first major comment still applies to trigrams; namely, the mere presence of trigrams does not entail that Merge was applied twice. Thus, in order to rule out this alternative interpretation the results, I strongly recommend the authors to test whether there exist constituent bigrams in order to create trigrams. If not, the authors can safely conclude that chimpanzees have the generative capacity to recombine diverse vocal sequences.

Reviewer #3 (Remarks to the Author):

The authors did good job in revising manuscript and to me, the current version is now clear and scientifically sounds. I think this paper has an important data on structural complexity in chimpanzee call sequences, which prompts future investigations on the origins of key linguistic features, such as compositionality and syntax.

We thank the reviewer for their positive feedbacks and for the final comments. Please find below the reviewers' comments in italic and our reply in bold.

Reviewers' comments:

Reviewer #1 (Remarks to the Author):

The authors have done a good job in addressing my comments in this revision of the manuscript. My critical point, that this analysis does not consider the meaning of the calls or call combinations, is now much clearer. I am happy to recommend publishing this paper, which I am sure will be well received by the community

We thank the reviewer for this recommendation.

Reviewer #2 (Remarks to the Author):

I thank the authors for addressing the comments.

The second of the two major comments in the first review was sufficiently addressed, by referring to the literature on human language acquisition. However, the first of the two major comments was not addressed, unfortunately. I perfectly understand that n-grams beyond trigrams cannot be analyzed in the paper due to data sparsity, but the alternative possibility suggested in the first major comment still applies to trigrams; namely, the mere presence of trigrams does not entail that Merge was applied twice. Thus, in order to rule out this alternative interpretation the results, I strongly recommend the authors to test whether there exist constituent bigrams in order to create trigrams. If not, the authors can safely conclude that chimpanzees have the generative capacity to recombine diverse vocal sequences.

We thank the reviewer for clarifying his suggestion. In our analysis we have characterised all the bigrams which are produced above chance as such (as bigrams alone), but also all the bigrams included in trigrams produced above chance. This provides us with the opportunity to assess exactly what the reviewer is suggesting. We found that most bigrams produced alone are reused into trigrams and could be constituent bigrams allowing to create trigrams (i.e. BK_SC, GR_PG, HO_PH, PH_PB, PH_PS, PS_SC and WH_PH). However, this does not apply to all the bigrams found in trigrams and some bigrams are only found in trigrams but not on their own (i.e. PB_BK, PB_PR, PB_PS, PG_PB, PS_PB and SC_PS). We added a description of these patterns in the discussion (Lines 655-667) as well as a figure to illustrate this partial overlap (Figure S9).

A second important point and a crucial requirement to test the merge idea is that we would need what A alone means, B alone means and C alone means, what AB means and what ABC means in order to assess the compositional power of the trigram ABC. This we want to exactly investigate in the future and is beyond the scope of the current study. We are therefore reluctant to make strong conclusions regarding the presence of a merge-like system in chimpanzees. We however include reference to how merge-like systems might be expected to look like for testing non-human generative capacity. "Contextual information will therefore help understanding the kind of combinatorial system beyond the generative capacity in chimpanzees (e.g., Rizzi 2016⁸⁹):

specifically, whether chimpanzees treat frequent bigrams as stored chunks to be reused in trigrams as a whole, or alternatively whether the transitional relationships between adjacent calls reflect simple adjunction or some more complex combinatorial mechanism beyond single combinations. “
lines 663-667.

Reviewer #3 (Remarks to the Author):

The authors did good job in revising manuscript and to me, the current version is now clear and scientifically sounds. I think this paper has an important data on structural complexity in chimpanzee call sequences, which prompts future investigations on the origins of key linguistic features, such as compositionality and syntax.

We thank the reviewer for this positive evaluation of our manuscript.